# MoVie: Multimodal Video Compression with Text Guidance

Jiaqi Hu [1]   Haoji Hu [1]   Heming Sun [2]   Lianrui Mu [1]

## Abstract

Most deep video codecs emphasize low-level motion modeling and remain largely semantics-agnostic, which can degrade perceptual quality in complex scenes. We propose **MoVie**, a **M**ultim**o**dal **Vi**d**e**o compression framework built on a Text-guided Video Transformer–CNN Mixed block (*Text-VideoTCM*). MoVie adopts a video-centric architecture that jointly models local spatial structures and temporal dynamics via window-based processing, delivering a favorable computation–perception trade-off. To incorporate semantics, we introduce dual-stage text fusion with *Extractor* and *Injector* modules. We further present history-conditioned coding that leverages both previous and aggregated historical frames, and a spatial–channel factorized entropy model that estimates probabilities over spatial neighborhoods and channel groups for adaptive bit allocation. Together, these designs reduce redundancy and improve rate control and temporal coherence, yielding reconstructions at low bitrates. On UVG and MCL-JCV, MoVie achieves $-$**50.23%** BD-rate for FID and $-$**14.64%** for LPIPS (VGGNet) relative to HM, while requiring only **55.76%** of DCVC-FM's per-pixel kMACs. A human perceptual study further confirms consistent subjective preference over strong baselines.

## 1. Introduction

Image and video compression are essential for efficient storage, transmission, and streaming in computer vision and multimedia applications. Traditional codecs like JPEG (Wallace, 1991), H.265/HEVC (Sullivan et al., 2012), and H.266/VVC (Bross et al., 2021) rely on hand-crafted mod-

---

[1]College of Information Science & Electronic Engineering, Zhejiang University, Hangzhou, China [2]School of Computing, Institute of Science Tokyo, Tokyo, Japan. Correspondence to: Haoji Hu <haoji_hu@zju.edu.cn>.

*Proceedings of the $43^{rd}$ International Conference on Machine Learning*, Seoul, South Korea. PMLR 306, 2026. Copyright 2026 by the author(s).

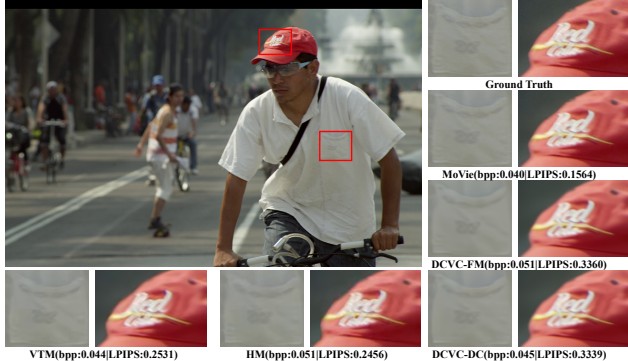

*Figure 1.* Visualization of the decoded 9th frame of the MCL-JCV videoSRC02 sequence using different methods. bpp denotes the bitrate, and LPIPS is a perceptual metric where lower is better.

ules such as block transforms, quantization, and entropy coding to exploit statistical redundancies.

With the rise of deep learning, video compression has made significant strides. The DVC framework (Lu et al., 2019) introduced end-to-end modeling of motion, residuals, and temporal priors, establishing a foundation for learned video coding. However, most existing models (Sheng et al., 2022; Li et al., 2023) still adopt frame-by-frame encoding or rely on low-level motion (e.g., optical flow), limiting their ability to model long-term temporal and semantic dependencies. While Transformer-based methods like VCT (Mentzer et al., 2022) offer improved temporal modeling, their use of global attention leads to prohibitive costs from the explosion of spatiotemporal tokens.

Although designed for video, many recent learned approaches still rely on image-based encoders, failing to fully exploit temporal structures. For example, VCT (Mentzer et al., 2022) uses the spatial ELIC backbone (He et al., 2022), and FLAVC (Zhang et al., 2025) employs LIC-TCM (Liu et al., 2023), lacking explicit modeling of inter-frame redundancy. Meanwhile, DCVC variants (Li et al., 2021; Sheng et al., 2022; Li et al., 2023) rely heavily on CNNs for motion and residual prediction, which limits the ability to capture long-range dependencies and semantics in dynamic scenes.

Parallel progress in multimodal learning shows natural-language descriptions, as revealed by CLIP (Radford et al., 2021), offer high-level semantics useful for vision tasks.

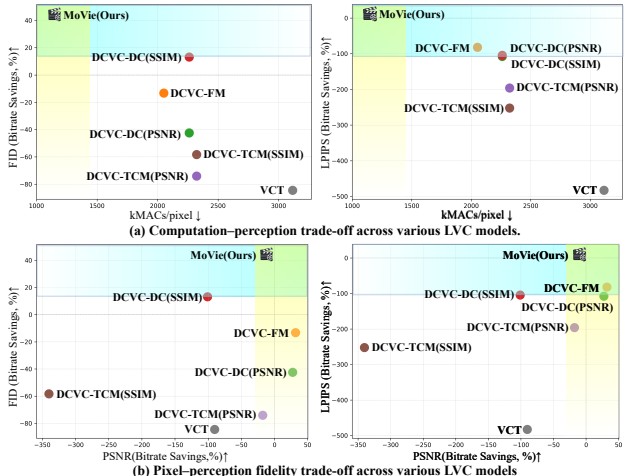

**(a) Computation–perception trade-off across various LVC models.**

**(b) Pixel–perception fidelity trade-off across various LVC models**

*Figure 2.* Trade-off performance of various LVC models in two aspects. Higher opacity indicates a better trade-off. We report bitrate savings (−BD-Rate, %) relative to HM-18.0 on UVG. Higher values indicate better performance. Note that LPIPS and FID are negated for BD-Rate calculation.

In image compression, integrating textual priors improves perceptual quality (Jiang et al., 2023; Qin et al., 2023; Lee et al., 2024). Although (Zhang et al., 2024) have achieved strong results in video compression with multimodal large models, their high memory usage and computational cost limit practical application.

To address these limitations, we propose a Text-guided Video Transformer–CNN Mixed block (*Text-VideoTCM*), a unified backbone that efficiently integrates textual semantics with localized spatiotemporal modeling. Our design couples a Video Swin Transformer (Liu et al., 2022) with a 3D CNN (Tran et al., 2015) to jointly capture local spatial and temporal dependencies. The Swin Transformer restricts attention to 3D windows, enabling efficient modeling of short-range motion dynamics, while the 3D CNN focuses on preserving local textures and motion boundaries. The resulting fused representation strikes an effective balance between semantic abstraction and detail preservation. To incorporate semantic guidance from text, we further introduce a two-stage fusion mechanism: *Extractor* and *Injector*. Specifically, we design an Extractor to apply cross-attention between video tokens and CLIP text features, generating frame-specific semantics, and an Injector to embed these cues into latent representations, enhancing semantic consistency during compression. As shown in Fig. 1, compared with other methods, our approach achieves satisfactory perceptual metrics and visual results.

Meanwhile, mainstream entropy models still treat latent features as short-range, missing richer dependencies spread across long sequences and channel groups. To bridge this gap, we design a *history-conditioned coding* strategy that

caches and reuses long-term latent summaries, enabling the Transformer to condition on historical frames. We further propose a spatial-channel factorized entropy model that estimates probabilities across spatial neighborhoods and channel groups, enabling adaptive bit allocation based on motion and semantic saliency. These modules improve rate control, reduce redundancy, and enhance temporal coherence.

Extensive experiments demonstrate that MoVie strikes an balance between perceptual quality, computational cost, and pixel-level fidelity. As shown in Fig. 2 (a), MoVie reduces computation by nearly half, owing to the efficient Text-VideoTCM block that models local spatiotemporal patterns. Our history-conditioned entropy model with spatial–channel factorization improves rate control by capturing long-range dependencies, while the text-guided fusion boosts perceptual quality by refining the distortion term. These components jointly improve overall BD-Rate performance, as shown in Fig. 2 (b). MoVie consistently outperforms perceptual baselines on perceptual metrics while maintaining strong PSNR performance.

Our key contributions are summarized as follows:

(1) We propose MoVie, a multimodal video compression framework that uses textual priors and cross-frame fusion for semantically guided and perceptually coherent compression.

(2) We propose the Text-VideoTCM block, which integrates textual semantics with localized spatiotemporal modeling to balance abstraction and detail preservation.

(3) We propose a history-conditioned coding strategy and a spatial–channel factorized entropy model to better capture long-term temporal dependencies and enhance entropy estimation.

(4) Extensive experiments demonstrate that MoVie achieves substantial BD-rate reductions while excelling in both computation–perception and pixel–perception fidelity trade-offs.

## 2. Related Work

### 2.1. CNN-based Video Compression

Early works like DVC (Lu et al., 2019) jointly optimize motion estimation, residual compression, and entropy modeling. Later methods adopt conditional coding (Liu et al., 2020) to enhance context representation, with DCVC (Li et al., 2021) introducing feature-domain motion compensation. DCVC-DC (Li et al., 2023) eliminates explicit motion estimation using deformable alignment with spatiotemporal priors, whereas DCVC-FM (Li et al., 2024b) adopts flow-matching for motion-free alignment. DCVC-RT (Jia et al.,

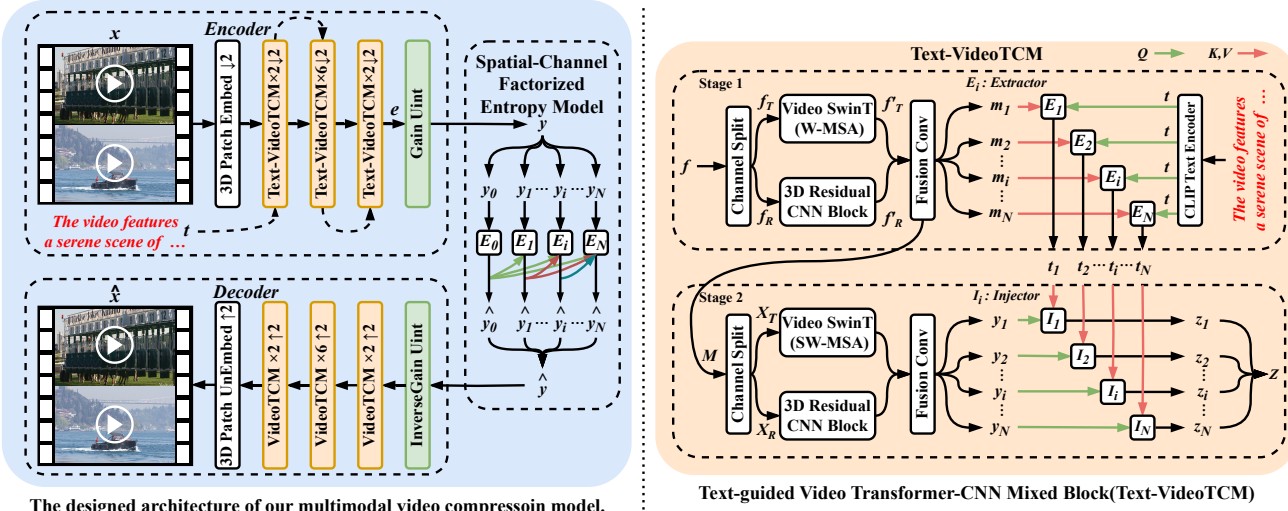

*Figure 3.* Overview of our multimodal video compression framework and the proposed *Text-guided Video Transformer–CNN Mixed Block* (*Text-VideoTCM*). The encoder extracts latent features using a sequence of *Text-VideoTCM* or *VideoTCM* blocks, where *3D Patch Embed/UnEmbed* modules perform spatiotemporal tokenization and reconstruction. Unlike *Text-VideoTCM*, *VideoTCM* does not use text guidance during decoding and lacks the *Extractor* (E) and *Injector* (I) modules.

2025) further pushes toward practical real-time neural video compression. Recent studies (Chen et al., 2024; Sheng et al., 2024) improve context diversity at the cost of increased complexity. Although effective at modeling temporal dependencies, 3D convolutions (Pessoa et al., 2020) are rarely used in CNN-based video compression due to their high cost.

## 2.2. Transformer-based Video Compression

VCT (Mentzer et al., 2022) leverages Transformer-based temporal modeling in the entropy coding module via implicit motion reasoning, achieving competitive compression with a simplified design, though at the cost of high memory overhead. Nonetheless, its autoencoder retains a CNN-based design for spatial representation learning. Based on VCT, FLAVC (Zhang et al., 2025) improves the encoder–decoder architecture using a Transformer-CNN mixed design (Liu et al., 2023). However, the autoencoder processes frames independently, neglecting temporal dependencies. In contrast, our method employs a video-specific autoencoder that fully exploits inter-frame consistency.

## 2.3. Text-guided Perceptual Compression

Following the success of vision-language models, text-guided image compression methods have recently emerged (Bhown et al., 2018; Weissman, 2023), leveraging high-level semantics to enhance perceptual quality. One line of work (Pan et al., 2022; Lei et al., 2023) explores using pre-trained text-guided generative models (e.g., diffusion models (Rombach et al., 2022)) as decoders. Some

approaches (Wan et al., 2025; Zhang et al., 2024) employ pretrained generative models or multimodal large language models for video compression, but this incurs substantial computational cost and storage overhead, making them impractical for efficient or large-scale deployment. Text-aware video autoencoders such as VideoVAE+ (Xing et al., 2025) are primarily designed for video generation and tokenization rather than rate–distortion or rate–perception optimized compression; their KL bottleneck and lack of entropy coding make direct comparison non-trivial (see Appendix M for a reconstruction-level comparison).

Another line of work trains decoders from scratch, where (Jiang et al., 2023) inject textual information into both encoder and decoder, while (Qin et al., 2023) introduce text only into the decoder via semantic-spatial aware blocks. Recent methods like TACO (Lee et al., 2024) show that adding text guidance only in the encoder can effectively boost perceptual quality without increasing compression cost. This opens a promising direction for end-to-end text-guided video compression, yet it remains unexplored in prior work. Our study bridges this gap by integrating efficient inter-frame modeling with semantic-aware compression into a unified, low-complexity framework.

## 3. Method

### 3.1. Overall Architecture

The overall architecture of our proposed multimodal video compression framework is illustrated in Fig. 3. Inspired by the success of Transformer-based models (Liu et al., 2022) in video understanding, we adapt the Video Swin

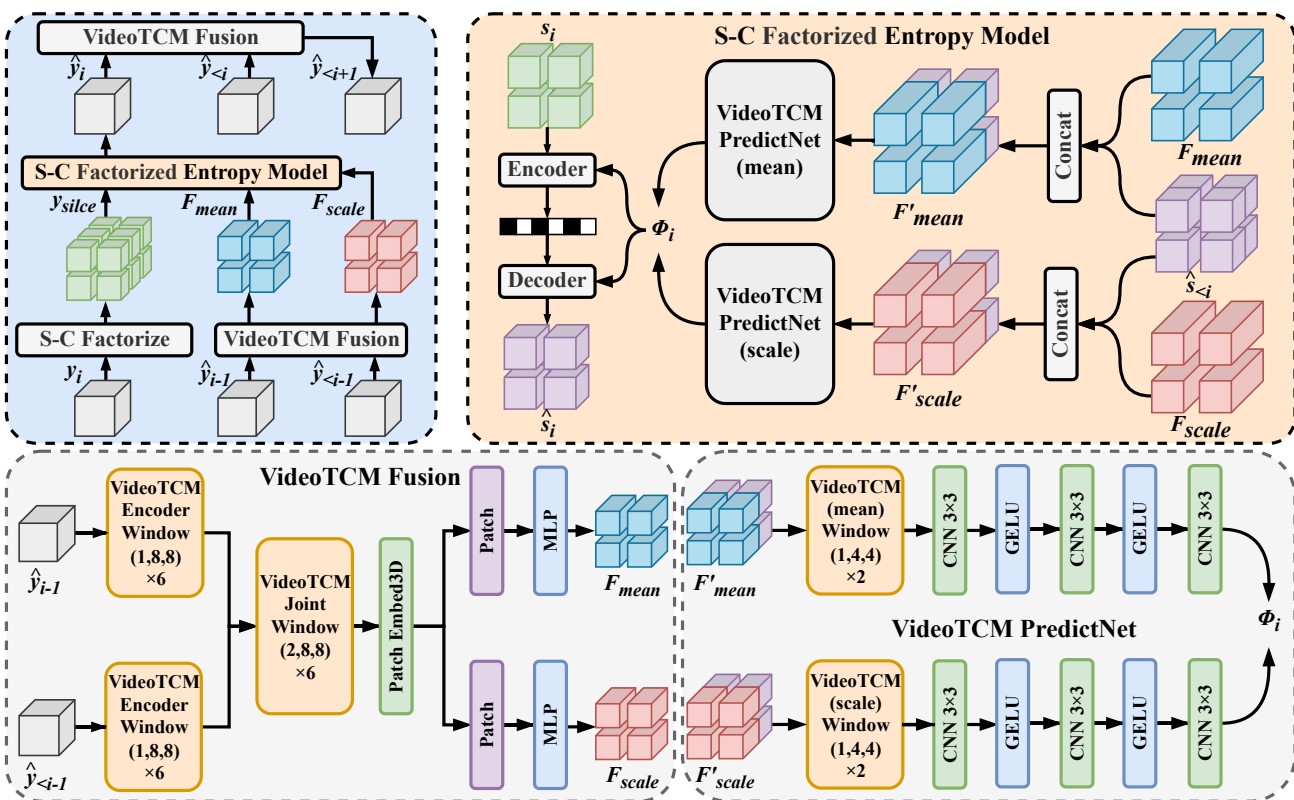

*Figure 4.* Overview of the proposed Spatial-Channel Factorized Entropy Model. Here, $y_i$ denotes the current encoded frame, while $\hat{y}_{i-1}$ and $\hat{y}_{<i-1}$ represent the previously and historical reconstructed frames. *S-C Factorized* refers to spatial-channel factorized.

Transformer—originally designed for recognition tasks—into a self-encoder structure optimized for video compression. Specifically, we augment the original architecture, which only contains a downsampling path, with a symmetric upsampling counterpart, forming a fully matched encoder-decoder design suitable for end-to-end video compression.

Our compression pipeline adopts a symmetric autoencoder architecture composed of an encoder $\mathbf{E}(\cdot)$ and a decoder $\mathbf{D}(\cdot)$, each consisting of three stages. In the encoder, we first partition the input video $x \in \mathbb{R}^{T \times H \times W \times 3}$ into non-overlapping patches along the spatial dimensions. Specifically, we divide the input into smaller spatial patches while preserving the full temporal resolution, unlike the original Video Swin Transformer which reduces the temporal dimension. This design choice ensures that every frame can be faithfully reconstructed in video compression tasks.

We then introduce a Text-VideoTCM block that captures spatial and temporal dependencies via parallel Video Swin Transformer (Liu et al., 2022) and 3D CNN (Tran et al., 2015) branches. The detailed architecture is described in the next section. Between stages, patch merging operations progressively reduce the spatial resolution while expanding the channel dimension, forming a hierarchical spatial pyramid. Notably, to ensure the reconstructability of compressed

features, we perform spatial downsampling only four times, resulting in a final feature resolution of $\frac{H}{16} \times \frac{W}{16}$.

To support one single model operating at multiple bit-rates, we follow AG-VAE (Cui et al., 2021) and adopt a gain-based rate-adjustment mechanism. Specifically, we incorporate a learnable Gain Unit in the encoder and its corresponding Inverse Gain Unit in the decoder. Given a user-specified rate-control index $s \in \{0, 1, \dots, S-1\}$—which maps to a preset Lagrangian multiplier $\lambda_s$—the units scale the latent features before quantization. We refer to this process as $f_{\text{gain}}$, with the detailed algorithm provided in Appendix C.

The latent representation is processed by an entropy model $\mathbf{H}(\cdot)$, which estimates the distribution of $y$ for accurate bitrate modeling. Before entropy coding, the features are quantized by $\mathbf{Q}(\cdot)$, which discretizes the continuous latent representations into a finite set of symbols. The resulting quantized representation $\hat{y}$ is then entropy coded into a bitstream and passed to the decoder to reconstruct the video $\hat{x}$. Thus, the overall pipeline of our framework can be summarized as follows:

$$e = \mathbf{E}(x,t), y = \mathbf{G}(s,e), \hat{y} = \mathbf{Q}(\mathbf{H}(y)), \hat{x} = \mathbf{D}(\hat{y}). \quad (1)$$

## 3.2. Text-Video TCM

In this section, we provide a detailed description of the proposed Text-VideoTCM block. This architecture employs window-based self-attention (W-MSA) and shifted window-based self-attention (SW-MSA) in two successive stages to model intra-window and inter-window dependencies, thereby achieving better local feature representations. Within each stage, a dual-branch design integrates a Video Swin Transformer module and a 3D Residual CNN module to jointly capture both local details and global structure, leading to enhanced visual performance. Furthermore, to improve perceptual quality, we incorporate text features in both stages using different fusion strategies, aiming to enhance semantic consistency and cross-modal alignment.

Assuming that the input tensor is $f \in \mathbb{R}^{T \times H_f \times W_f \times C}$, we first evenly split it along the channel dimension into two sub-tensors, $f_T$ and $f_R$, each with a shape of $\mathbb{R}^{T \times H_f \times W_f \times \frac{C}{2}}$. Then, $f_T$ and $f_R$ are respectively fed into a Video Swin Transformer(Video SwinT) block and a 3D Residual CNN Block, which are used to extract local and non-local features in parallel, resulting in the outputs $f'_T$ and $f'_R$. Afterward, these two features are fused via a residual convolution module to obtain the final representation $M \in \mathbb{R}^{T \times H_f \times W_f \times C}$, and can be expressed as $\{m_1, m_2, \ldots, m_i, \ldots, m_T\} \in \mathbb{R}^{H_f \times W_f \times C}$. The entire fusion procedure is defined as:

$$
\begin{aligned}
f_T, f_R &= \mathrm{Split}(f), \\
f'_T = \mathrm{VideoSwinT}(f_T), f'_R &= \mathrm{Res3D}(f_R), \\
M &= f + \mathrm{Conv3D}_{1 \times 1}(\mathrm{Cat}(f'_T, \ f'_R)).
\end{aligned} \quad (2)
$$

The textual description is encoded by a frozen CLIP text encoder into $t \in \mathbb{R}^{L \times 512}$, where we fix the token length to $L = 38$ by truncating or padding the CLIP token sequence. Since it describes the entire video, $t$ is tiled $T$ times along the temporal axis and fused with image features $X$. Extractors $\{E_1, \ldots, E_T\}$ then apply cross-attention to each token $x_i$ with $t$: following CLIP, $t$ serves as query $Q$ and $m_i$ provides keys/values $(K, V)$. This yields per-frame text features $\{t_i\}_{i=1}^{T} \in \mathbb{R}^{T \times L \times 512}$. The output of the extractor for the $i$-th frame is computed as:

$$
t_i = E_i(t, m_i) = t + \mathrm{CrossAttention}(t, \mathrm{Lin}(m_i)). \quad (3)
$$

Here, $\mathrm{Lin}(\cdot)$ projects image features to the text feature dimension, and $\mathrm{CrossAttention}(\cdot)$ is a multi-head attention module with text as $Q$, and image as $K, V$.

Stage 2 re-encodes the feature map $X$ with the same dual-branch design as Stage 1; the Video Swin Transformer uses shifted-window attention. To inject per-frame semantics into the visual stream, we introduce injectors $\{I_1, \ldots, I_T\}$. Each $I_i$ takes the second-stage image feature $y_i$ and its paired text-guided token $t_i$, producing $z_i$ as:

$$
z_i = I_i(y_i, t_i) = y_i + \mathrm{CrossAttention}(y_i, \mathrm{Lin}(t_i)). \quad (4)
$$

The image feature serves as $Q$, and the text feature as $K, V$ in the cross-attention module.

## 3.3. Spatial–Channel Factorized Entropy Model

Fig. 4 presents an overview of our proposed Spatial–Channel Factorized Entropy Model, which aims to enhance entropy estimation by jointly capturing spatial, temporal, and channel-wise dependencies in video latents. For clarity, we omit the batch and temporal indices in the following derivation and focus on a single frame latent $y_i \in \mathbb{R}^{H \times W \times C}$.

**Left: Windowed Spatial–Channel Factorization and Temporal Cache.** We first partition $y_i$ into non-overlapping $8 \times 8$ spatial windows and denote the total number of windows by $N_w = (H/8)(W/8)$. Each window is then split along the channel dimension into 6 equal-sized groups:

$$
y_i[p, q] = \left(y_i^{(1)}[p, q], \ldots, y_i^{(6)}[p, q]\right) \in \mathbb{R}^{8 \times 8 \times C}. \quad (5)
$$

We call $y_i^{(g)}[p, q]$ a *channel slice* and index it by the triple $(p, q, g)$. In the right part of Fig. 4, each slice is denoted by $s_i$ and is processed sequentially along the channel dimension, i.e., at step $g$ the input to the S–C Factorized Entropy Model encoder is $s_i = y_i^{(g)}[p, q]$. The corresponding decoded slice $\hat{s}_i$ is then stored and concatenated with all previously decoded slices in the same window to form $\hat{s}_{<i}$, which serves as an additional conditional input when predicting the distribution of the next channel slice.

To effectively model temporal priors, we employ a VideoTCM Fusion module that aggregates information from the previous latent feature $\hat{y}_{i-1}$ and historical context latent feature $\hat{y}_{<i-1}$ to predict the mean $F_{\mathrm{mean}}$ and scale $F_{\mathrm{scale}}$ parameters for the current latent $y_i$.

Then, $y_{\mathrm{slice}}$, along with the predicted $F_{\mathrm{mean}}$ and $F_{\mathrm{scale}}$, are fed into the spatial-channel entropy coding block to produce the quantized latent $\hat{y}_i$. The output is further $\hat{y}_i$ fused with $\hat{y}_{<i}$ to generate $\hat{y}_{<i+1}$ for the entropy modeling of $y_{i+1}$.

The recursive modeling process can be expressed as follows:

$$
\begin{aligned}
p_\theta\left(y_i \mid \hat{y}_{i-1}, \hat{y}_{<i-1}\right) &= \mathrm{Entropy}_\theta\left(\hat{y}_{i-1}, \hat{y}_{<i-1}\right), \\
\hat{y}_{<i} = f\left(\hat{y}_{<i-1}, \hat{y}_{i-1}\right), &\quad \hat{y}_{<1} = y_0.
\end{aligned} \quad (6)
$$

**Right: S-C Factorized Entropy Model.** The right part of the figure illustrates the proposed spatial-channel factorized entropy model, which refines the entropy parameters by learning spatial- and channel-aware distributions.

From the left part of the pipeline, the obtained latent tensor $y_{\mathrm{slice}}$ is first split along the channel dimension into a set of features $s_i$. Each channel slice $s_i$ is then entropy coded sequentially to produce the quantized output $\hat{s}_i$.

Meanwhile, the conditional inputs $F_{\mathrm{mean}}$ and $F_{\mathrm{scale}}$ are concatenated with the encoded results of previous channels $s_{<i}$,

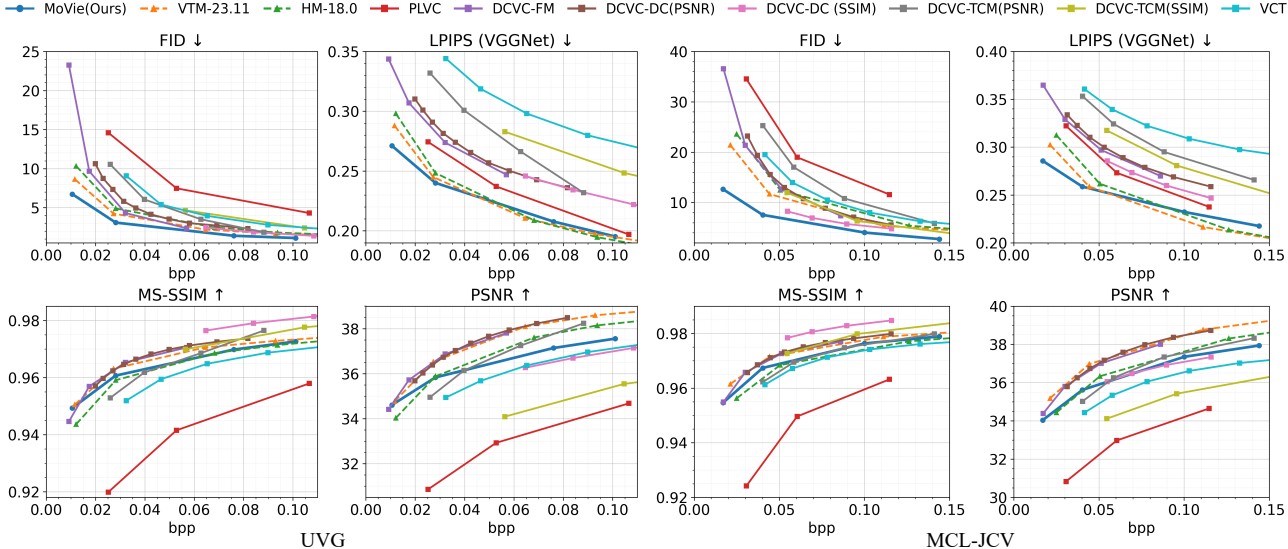

*Figure 5.* Overall compression performance with RD curves (↓ lower is better, ↑ higher is better).

yielding updated representations $F'_{\text{mean}}$ and $F'_{\text{scale}}$. These are then fed into two independent prediction networks (predict-Net) to produce the quantization parameters $\phi_i$.

## 4. Experiments

### 4.1. Experimental Setup

We optimize the model with four losses over the original video $x$, textual description $t$, reconstructed output $\hat{x}$, and quantized latent $\hat{y}$, the overall loss is defined as:

$$
\mathcal{L} = \sum_{i=1}^{T} \Big\{ \mathcal{R}(\hat{y}_i) + \lambda_s \mathcal{D}(x_i, \hat{x}_i) + k_p \text{LPIPS}(x_i, \hat{x}_i) \\
+ k_j \big[ \mathcal{L}_{\text{con}}(f_I(\hat{x}_i), f_T(t)) + \beta \| f_I(x_i) - f_I(\hat{x}_i) \|_2^2 \big] \Big\}.
\tag{7}
$$

The first three terms correspond to standard losses: bitrate $\mathcal{R}(\cdot)$, distortion $\mathcal{D}(\cdot, \cdot)$ (MSE), and perceptual similarity (LPIPS). $f_I(\cdot)$ and $f_T(\cdot)$ denote CLIP's image and text encoders, and $\mathcal{L}_{\text{con}}$ is the CLIP contrastive loss. We adopt the hyperparameters from TACO (Lee et al., 2024), setting $k_p=1$, $k_j=0.005$, and $\beta=40$. Specifically, we follow the same schedule as CompressAI (Bégaint et al., 2020), assigning $\lambda_s = 0.00045, 0.0018, 0.0067,$ and $0.0200$ for $s = 0, 1, 2, 3$, respectively. Following VCT (Mentzer et al., 2022), we adopt a three-stage training: autoencoder pre-training with distortion/perceptual losses(w/o $\mathcal{R}$), entropy model training with $\mathcal{R}$, and joint fine-tuning. We train the three stages for 1M, 250k, and 500k steps respectively on two NVIDIA A6000 GPUs.

### 4.2. Datasets and Evaluation

Following standard practice, we train our model on Vimeo90K (Xue et al., 2019) using random $256 \times 256$ crops. For evaluation, we adopt UVG (Mercat et al., 2020) and MCL-JCV (Wang et al., 2016) to ensure scene diversity. Additional results on four HEVC datasets are provided in the Appendix N. All captions are generated by LLaVA (Li et al., 2024a), with prompts included in the Appendix F. We evaluate the first 96 frames of each sequence under the standard protocol. Rate-distortion (RD) performance is assessed using five metrics: FID, FVD, and LPIPS for perceptual quality, and PSNR and MS-SSIM for pixel-level fidelity. We additionally report WarpError (Teed & Deng, 2020) for temporal coherence evaluation.

### 4.3. Rate-Distortion Performance

Our baselines include the Transformer model VCT (Mentzer et al., 2022) and the DCVC family—DCVC-TCM (Sheng et al., 2022), DCVC-DC (Li et al., 2023), and DCVC-FM (Li et al., 2024b). For DCVC-TCM/DCVC-DC we report both PSNR- and MS-SSIM-optimized variants; DCVC-FM is reported in its unified configuration. We also include traditional reference codecs HM-18.0 and VTM-23.11, using their latest releases; encoding commands are provided in Appendix D. Additionally, we compare with PLVC (Yang et al., 2022), a representative work in perceptual video compression. A separate comparison with DCVC-RT (Jia et al., 2025) is provided in Section L.

As shown by the RD curves in Fig. 5 and the BD-Rate results in Table 1, MoVie delivers state-of-the-art perceptual quality, especially at low bitrates. It consistently attains

| Method | kMAC/pixel | UVG | | | | MCL-JCV | | | |
|---|---|---|---|---|---|---|---|---|---|
| | | FID | LPIPS | MS-SSIM | PSNR | FID | LPIPS | MS-SSIM | PSNR |
| HM-18.0 | / | 0 | 0 | 0 | 0 | 0 | 0 | 0 | 0 |
| VTM-23.11 | / | -17.38 | -1.51 | -23.93 | -29.14 | -17.03 | -9.52 | -30.00 | **-30.08** |
| PLVC | / | 219.19 | 41.19 | 356.35 | 565.80 | 98.93 | 42.90 | 229.91 | 323.97 |
| DCVC-FM | 2051.8 | 13.21 | 82.00 | -29.49 | **-31.82** | 5.84 | 74.77 | -28.50 | -25.82 |
| DCVC-DC (PSNR) | 2260.8 | 42.42 | 107.60 | -30.25 | -27.30 | 12.70 | 93.48 | -31.70 | -26.58 |
| DCVC-DC (SSIM) | 2260.8 | -13.11 | 104.20 | **-62.36** | 101.10 | -26.82 | 68.00 | **-66.43** | 33.12 |
| DCVC-TCM (PSNR) | 2322.1 | 74.03 | 195.94 | -1.56 | 17.80 | 56.96 | 187.30 | 0.60 | 17.54 |
| DCVC-TCM (SSIM) | 2322.1 | 58.28 | 252.23 | -40.23 | 340.24 | -13.51 | 141.26 | -45.13 | 182.66 |
| VCT | 3115.9 | 84.43 | 482.72 | 47.61 | 90.23 | 29.66 | 349.99 | 23.96 | 79.13 |
| **MoVie (Ours)** | **1144.1** | **-45.09** | **-13.03** | -13.98 | 7.62 | **-55.36** | **-16.25** | -15.67 | 10.91 |

*Table 1.* Comparison on UVG and MCL-JCV in BD-Rate (%) with HM-18.0 as anchor. BD-Rate is computed as $-M$ for lower-is-better metrics. MAC is measured in kMACs/pixel.

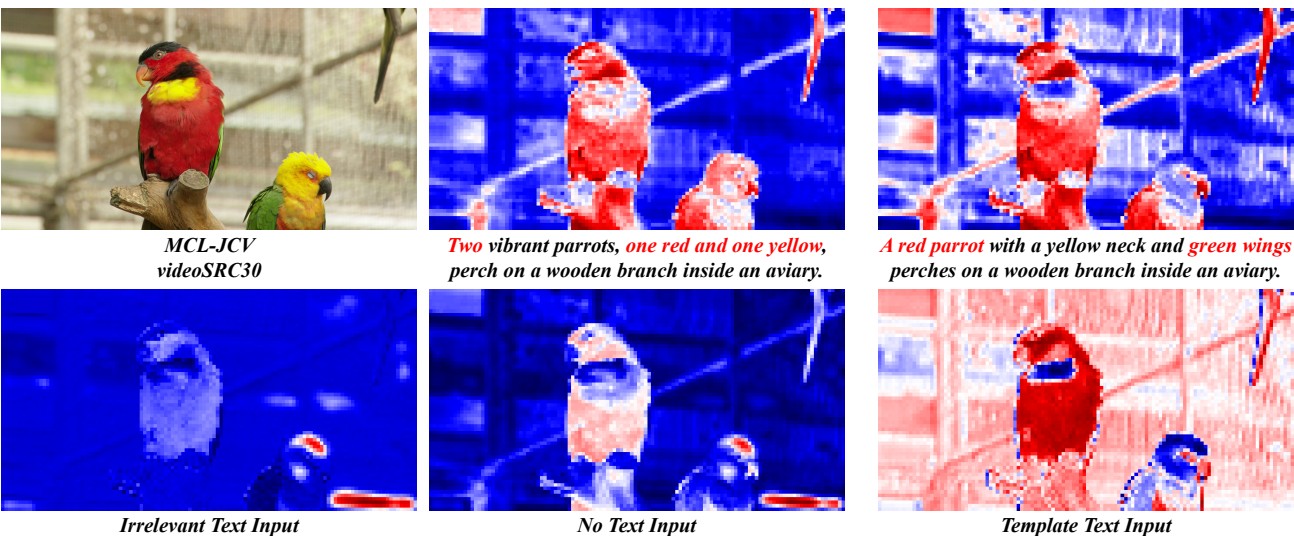

*Figure 6.* Visualization comparison of the stacked top 8 highest-entropy channels of encoder output $y$ under different text inputs.

the lowest FID across the full rate range, indicating that its text-guided, semantics-aware features better align with natural image statistics. Moreover, on LPIPS, MoVie leads at low bitrates and achieves the largest BD-Rate reductions.

Although MoVie is not the top performer in PSNR/MS-SSIM, it remains competitive with HM in PSNR BD-Rate and reduces MS-SSIM BD-Rate by 13.98%—despite not being explicitly optimized for SSIM. In the low-bitrate regime (leftmost points in Fig. 5), it achieves comparable fidelity to other methods. Compared with PSNR-oriented models, MoVie offers better perceptual quality (lower LPIPS) with only a minor PSNR trade-off. While MS-SSIM–optimized models excel on MS-SSIM, MoVie significantly outperforms them on the three metrics.

In addition, Table 3 reports separate encoding and decoding times together with the encoder/decoder complexities in kMACs/pixel. MoVie attains both the fastest codec runtime and the lowest computational cost, largely thanks to the efficiency of its window-based processing design. Including

the LLaVA text-generation overhead (132 ms per frame), the total encoding time of MoVie (456 ms) remains comparable to DCVC-DC (458 ms) and DCVC-FM (439 ms).

### 4.4. Effect of Text Guidance

To better isolate the effect of text guidance, Table 2 reports variants with different text-injection locations: Enc-V/Dec-V (Vie, no text on either side), Enc-T/Dec-V (MoVie, text only in the encoder), Enc-V/Dec-T (text only in the decoder), and Enc-T/Dec-T (text on both encoder and decoder). The results show that MoVie (Enc-T/Dec-V) achieves the best perceptual scores (FID and LPIPS) with only a minor drop in PSNR compared to Vie. In contrast, adding text only to the decoder (Enc-V/Dec-T) or to both encoder and decoder (Enc-T/Dec-T) does not lead to further improvements and can even hurt reconstruction quality.

A plausible explanation is that, during autoencoder training, the encoder determines what information is preserved in

| Method | kMACs/pixel | Inference time | PSNR↑ | MS-SSIM↑ | FID↓ | LPIPS↓ |
|---|---|---|---|---|---|---|
| Enc-V / Dec-V (Vie) | **226.07** | 187ms | **46.78** | **0.99826** | 0.0062 | 0.0018 |
| **Enc-T / Dec-V (MoVie)** | 235.12 | 197ms | 46.49 | 0.99796 | **0.0041** | **0.0009** |
| Enc-V / Dec-T | 235.12 | 196ms | 45.08 | 0.99673 | 0.0053 | 0.0012 |
| Enc-T / Dec-T | 244.11 | 206ms | 45.69 | 0.99686 | 0.0057 | 0.0012 |
| LIC-TCM | 944.50 | 577ms | 45.58 | 0.99757 | 0.0065 | 0.0022 |
| ELIC | 479.52 | **183ms** | 43.01 | 0.99411 | 0.0221 | 0.0098 |
| Global Text | 228.76 | 190ms | 40.04 | 0.99565 | 1.6403 | 0.0037 |
| Copy Text | 230.76 | 190ms | 44.59 | 0.99789 | 0.0212 | 0.0020 |
| Constant Text | 235.12 | 197ms | 40.93 | 0.99138 | 0.0593 | 0.0111 |

*Table 2.* Comparison of different autoencoders on UVG, **without** entropy coding. The reported inference time corresponds to the average time to reconstruct a single frame. All measurements are obtained on an NVIDIA RTX 3090 GPU.

| Method | $T_{enc}$ | $T_{text}$ | $T_{total}$ | $T_{dec}$ | kMAC/px |
|---|---|---|---|---|---|
| DCVC-DC | 458 | – | 458 | 404 | 2260.8 |
| DCVC-FM | 439 | – | 439 | 377 | 2051.8 |
| VCT | 614 | – | 614 | 531 | 3115.9 |
| MoVie | **324** | 132 | 456 | **255** | **1144.1** |

*Table 3.* Per-frame time (ms) and complexity comparison. Text denotes LLaVA caption generation overhead. All times averaged on an NVIDIA A6000 GPU.

the latent space. When text is injected at the encoder side, the latent representation is shaped to align with semantic cues, so the decoder can reconstruct text-relevant structures more faithfully from a coherent latent. If text is introduced only at the decoder, the latent remains video-only and the decoder must "correct" or hallucinate details from insufficient features, which can conflict with the latent and degrade both fidelity and perceptual quality. Injecting text on both sides over-conditions the model with similar guidance twice, making optimization harder without providing additional semantic information beyond the encoder-side guidance, which explains why Enc-T/Dec-T does not outperform the simpler Enc-T/Dec-V design.

In addition, we include two widely used *image*-compression backbones as autoencoder variants for video: LIC-TCM (Liu et al., 2023) and ELIC (He et al., 2022). These baselines illustrate the trade-off between reconstruction quality and efficiency in image-oriented designs. LIC-TCM delivers relatively strong reconstruction quality but incurs very high computational cost and long inference time, whereas ELIC is lightweight and fast but suffers from noticeably worse reconstruction quality. By contrast, our video-specific encoder–decoder is designed to explicitly exploit inter-frame relationships, which allows it to achieve both lower complexity and better reconstruction quality on video.

We also compare different text-integration strategies. As shown in Table 2, *Global Text* uses a single clip-level caption for all frames, and *Copy Text* simply replicates this caption at each time step. We also include a *Constant Text* variant, which keeps the MoVie-style architecture but replaces captions with the same fixed sentence across clips and frames to isolate meaningful text.

The results show that naively sharing one caption across frames is ineffective. Both Global Text (single-stage fusion with a clip-level embedding) and Constant Text (non-informative, fixed captions) exhibit clear drops in distortion and perceptual metrics, indicating that frame- or content-agnostic descriptions cannot match diverse frame content. Copy Text helps slightly but still lags behind our two-stage design. By first adapting the clip-level caption into *frame-specific* text features and then fusing them temporally, our approach exploits informative, content-dependent text and better captures frame-to-frame variations, yielding higher perceptual quality at similar complexity.

| Text | bpp | PSNR↑ | MS-SSIM↑ | FID↓ | LPIPS↓ |
|---|---|---|---|---|---|
| Ground Truth | **0.011** | **34.61** | **0.94930** | **6.731** | **0.19388** |
| Rephrased Text | 0.011 | 34.57 | 0.94928 | 6.731 | 0.19411 |
| Irrelevant Text | 0.035 | 29.35 | 0.92912 | 15.129 | 0.21280 |
| Without Text | 0.032 | 29.77 | 0.93068 | 14.694 | 0.21164 |
| Template Text | 0.011 | 34.26 | 0.93934 | 11.657 | 0.20340 |

*Table 4.* Robustness to textual perturbations. All results use the same model; bpp variations arise only from input text differences.

### 4.5. Stability Evaluation

To assess the robustness of semantic guidance, we evaluate different textual inputs for the same video (Table 4). We use a factual caption as Ground Truth and GPT (Hurst et al., 2024) to generate semantically similar alternatives (Rephrased Text). We also test two challenging settings: irrelevant descriptions (Irrelevant Text) and no descriptions (Without Text), where a simple fallback template is used as a generic substitute (Template Text).

As shown in Table 4, our model remains stable with semantically aligned text, showing robustness to paraphrasing. In contrast, irrelevant or missing text significantly degrades perceptual and fidelity performance, highlighting the importance of semantic guidance. Our template serves as a fallback when accurate captions are unavailable, yielding better reconstructions than incorrect or missing text, though still worse than accurate descriptions. The specific template is provided in Appendix I.

As shown in Fig. 6, we visualize the stacked top-8 highest-entropy channels under different text inputs. When the description covers both birds, the model attends well to both targets; with a single-bird description, the other bird receives little attention. For irrelevant or no-text inputs, the attention distribution becomes diffuse and lacks focus. Template text provides some emphasis, but the contrast with the background remains limited.

### 4.6. Temporal Coherence Evaluation

To quantitatively validate the temporal coherence claim, we adopt Mean Warping Error (Teed & Deng, 2020) as a dedicated metric. We estimate optical flow between adjacent reconstructed frames using pretrained RAFT, warp the frame at time $t$ toward $t+1$, and compute the mean absolute pixel error. A lower value indicates better frame-to-frame stability.

Since different methods operate at different bitrates, we evaluate each method across four rate points and compute BD-Rate with WarpError as the quality axis, using HM as the anchor:

| Method | BD-Rate (WarpError $\downarrow$) |
|---|---|
| HM (anchor) | 0% |
| VTM | −14.42% |
| DCVC-FM | +3.19% |
| **MoVie (Ours)** | **−15.88%** |

Table 5. Temporal coherence comparison via BD-Rate on WarpError (UVG). A negative value indicates better temporal stability at lower bitrate relative to HM.

MoVie achieves the best WarpError BD-Rate (−15.88%), outperforming VTM (−14.42%) and DCVC-FM (+3.19%). This confirms that text-guided semantic conditioning helps maintain consistent high-level content across frames and reduces perceptual temporal fluctuations.

To further examine long-horizon stability, we additionally compress 384-frame sequences (4× longer than the standard 96-frame evaluation) and break down WarpError BD-Rate into four 96-frame segments. The results show no progressive error accumulation: MoVie consistently outperforms HM across all segments (f001–096: −51.66%, f097–192: −81.94%, f193–288: −74.99%, f289–384: −71.91%).

### 4.7. Component Ablation

To isolate the contribution of each design component, we conduct a progressive ablation by retraining models with different module combinations. Starting from the baseline (VCT backbone), we progressively add the S-C entropy model, VideoTCM, and the text-guided module:

The results show complementary gains from all components. The S-C entropy model improves all three metrics.

| Setting | BD-Rate (LPIPS) | BD-Rate (FID) | BD-Rate (PSNR) |
|---|---|---|---|
| Baseline (anchor) | 0.00% | 0.00% | 0.00% |
| +S-C Entropy Model | −45.39% | −52.04% | −5.70% |
| +VideoTCM | −73.60% | −71.38% | −19.51% |
| +Text-Guided (Full) | **−87.07%** | **−81.94%** | −16.07% |

Table 6. Progressive ablation on UVG. Each row adds one component to the previous setting. All variants are retrained.

VideoTCM yields the best PSNR BD-Rate (−19.51%), confirming its benefit for pixel-level fidelity. The text-guided module provides the strongest perceptual gains, reaching −87.07% in LPIPS and −81.94% in FID.

### 4.8. Human Perceptual Study

To provide direct human-subjective evidence beyond automated metrics, we conducted a perceptual study with 20 naive subjects evaluating 20 videos from four datasets (UVG, MCL-JCV, HEVC-C, HEVC-D) at low bitrates (bpp ≈ 0.01). In each trial, subjects viewed the reference video alongside our result and a baseline, and chose the one perceptually closer to the reference.

| Baseline | Win Rate | 95% CI | BT Score |
|---|---|---|---|
| DCVC-FM | 73.75% | [69.1%, 77.9%] | −1.033 |
| VTM | 78.00% | [73.7%, 81.8%] | −1.266 |
| PLVC | 85.50% | [81.7%, 88.6%] | −1.774 |

Table 7. Human perceptual study results. Win Rate indicates the percentage of trials where MoVie is preferred. BT Score is the Bradley–Terry score (MoVie fixed at 0; lower baseline scores indicate weaker preference).

MoVie is consistently preferred over all baselines, with win rates of 73.75%, 78.00%, and 85.50% against DCVC-FM, VTM, and PLVC, respectively. This provides direct evidence that semantic guidance improves perceived quality beyond what automated metrics capture.

## 5. Conclusion

We propose MoVie, a multimodal video compression framework that embeds textual semantics into a unified Video Transformer–CNN block. Through joint spatiotemporal modeling and two-stage text fusion, MoVie improves perceptual quality while reducing computation. A spatial–channel factorized entropy model and history-conditioned coding further enhance efficiency. Extensive experiments, including quantitative temporal coherence evaluation and a human perceptual study, demonstrate that MoVie achieves an excellent trade-off between fidelity and perceptual quality, particularly at low bitrates, highlighting the potential of multimodal guidance. However, its performance is sensitive to text accuracy, and future work will focus on more robust semantic guidance and exploring video-specific vision-language encoders to ensure stable compression under imperfect text conditions.

## Impact Statement

This work contributes to efficient video compression by using textual guidance to improve perceptual quality at low bitrates. It may help reduce storage and bandwidth requirements for video streaming, communication, and multimedia applications, especially when computational resources are limited.

Potential risks mainly come from the use of text descriptions. Inaccurate or biased text may affect reconstruction quality or guide the model toward undesirable visual emphasis. More broadly, improved video compression could be used in large-scale video systems where privacy and responsible data handling should be carefully considered. This work does not involve new data collection or surveillance applications, and all experiments are conducted on public research datasets.

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

## A. Appendix

The appendix provides detailed descriptions of the network architecture, training strategy, dataset usage, prompt construction, additional experimental results, comparisons under various traditional codec settings, and the command-line configurations used for HM and VTM baselines.

## B. Dataset

We conduct training on the Vimeo-90k dataset (Xue et al., 2019), which is distributed under the MIT License. For evaluation, we use three publicly available and widely adopted datasets: the UVG dataset (Mercat et al., 2020) (BY-NC License), the MCL-JCV dataset (Wang et al., 2016) (copyright details are available online), and the standard HEVC test sequences (Bossen, 2010; Flynn et al., 2015). All datasets are intended for academic research purposes and can be freely accessed online. Furthermore, we have manually verified that none of the data contains personally identifiable information or offensive content.

In the main paper, we report results on the UVG and MCL-JCV datasets. Specifically, Fig. 8 and Table 13 present the rate-distortion curves and BD-rate results for the HEVC-B and HEVC-C datasets, while Fig. 9 and Table 14 show the corresponding results on the HEVC-D and HEVC-E datasets. We further present qualitative comparisons in Fig. 10 and Fig. 11 across different datasets. As shown in Fig. 10, our method achieves better reconstruction in high-frequency regions, preserving textures more faithfully. Moreover, Fig. 11 demonstrates that our method maintains visually pleasing reconstruction quality even when operating at only half the bitrate of other methods.

---

**Algorithm 1** GainUnit: Scale-conditioned Modulation

---

1: **function** GAIN INIT$(x, s)$
2: $\quad g \leftarrow \exp(\log g_s)$
3: $\quad G \leftarrow \mathrm{reshape}(g, [1, 1, C, 1, 1])$
4: $\quad$ **return** $x \cdot G$
5: **end function**
6: **function** INVERSE GAIN INIT$(x, s)$
7: $\quad g \leftarrow \exp(-\log g_s)$
8: $\quad G \leftarrow \mathrm{reshape}(g, [1, 1, C, 1, 1])$
9: $\quad$ **return** $x \cdot G$
10: **end function**

---

## C. Rate Control

We implement a simple yet lightweight rate control mechanism by introducing a learnable Gain Unit and Inverse Gain Unit as an additional modulation layer. The complete algorithm is provided in Algorithm 1. We emphasize that this is a preliminary and straightforward implementation. Our modular codec design allows for more advanced rate control strategies, which we leave for future exploration.

Here, $\log g_s \in \mathbb{R}^C$ denotes a learnable gain vector indexed by scale $s$, and $x \in \mathbb{R}^{B \times D \times C \times H \times W}$. The gain is broadcast over spatial and temporal dimensions after reshaping to $G \in \mathbb{R}^{1 \times 1 \times C \times 1 \times 1}$.

## D. Traditional Codec

We used the official reference implementations of HM and VTM for traditional video compression:

- **HM**: version **18.0**, compiled on *Linux (GCC 13.2.0, 64-bit)* with *Range Extensions (RExt)* enabled.

- **VTM**: version **23.11**, compiled on *Linux (GCC 13.2.0, 64-bit)* with *AVX2 SIMD* acceleration.

These implementations are widely adopted as standard benchmarks for evaluating learned video compression frameworks, and all experiments were conducted under consistent compilation environments to ensure fair comparison.

We followed the *common test conditions (CTC)* specified by the JVET and JCT-VC standardization groups to ensure fair and reproducible comparison. The encoder configurations used for traditional codecs are as follows:

- **HM**:
  We used the `encoder_randomaccess_main.cfg` configuration file from HM-18.0.

- **VTM**:
  We used the `encoder_randomaccess_vtm.cfg` configuration file from VTM-23.11.

Each video was encoded with the following general command-line options:

```
EncoderAppStatic \
  -c {config file}
  --InputFile={input file}
  --InputBitDepth=8
  --OutputBitDepth=8
  --FrameRate={frame rate}
  --FramesToBeEncoded={frame number}
  --SourceWidth={width} --SourceHeight={height}
  --QP={qp}
```

## E. Training Strategy

We adopt a three-stage training strategy for efficiency and stability.

- **Stage I:** Only the autoencoder is trained without the bitrate term, using a small number of frames to accelerate early convergence.

- **Stage II:** The entropy model is introduced and trained with all loss terms, using longer sequences for improved temporal modeling.

- **Stage III:** The entire model is jointly optimized with a reduced learning rate.

This strategy allows gradual integration of rate modeling while maintaining training stability.

| | Components trained | Loss | $B$ | $N_F$ | LR | Steps |
|---|---|---|---|---|---|---|
| Stage I | Autoencoder | w/o $r$ | 4 | 1 | 2E−4 | 1M |
| Stage II | Entropy Model | all | 4 | 7 | 2E−4 | 250k |
| Stage III | All Components | all | 4 | 7 | 5E−5 | 500k |

*Table 8.* We split training in three stages for training efficiency. $r$ is bitrate, $d$ is distortion, $B$ is batch size, $N_F$ the number of frames.

## F. Prompt Construction

We adopt LLaVA-Video (Li et al., 2024a) (Qwen2 version) as our video-text generation backbone. For each short video clip from video dataset, we uniformly sample 64 frames and construct the following prompt:

```
The video lasts for X.XX seconds, and N frames are uniformly sampled from
it.  These frames are located at T₁s, ..., T_Ns.
Please answer the following questions related to this video.
Please describe this video in detail.
```

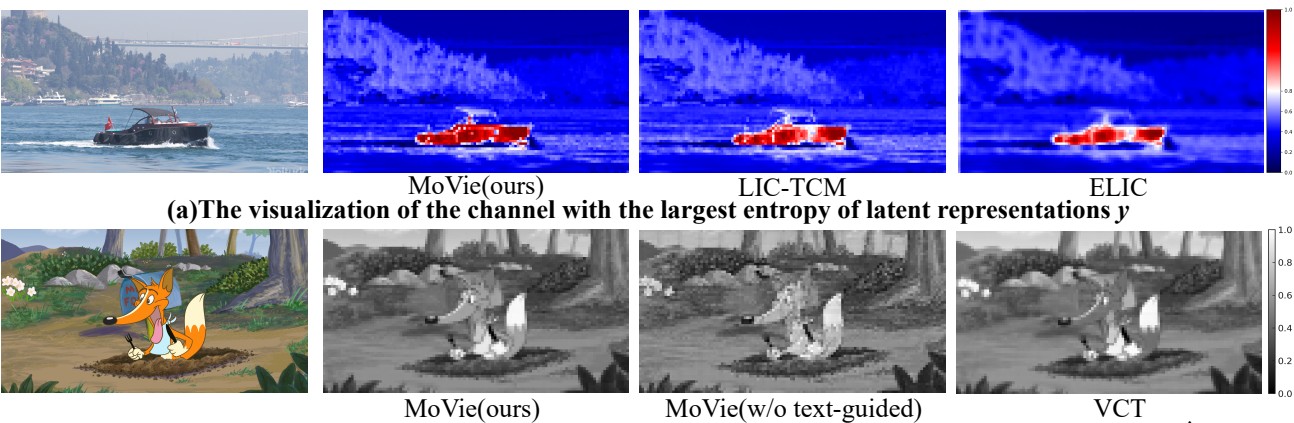

(a)The visualization of the channel with the largest entropy of latent representations $y$

**(b) The visualization of the channel with the largest entropy of compressed latent representations $\hat{y}$**

*Figure 7.* Visual comparison of different methods.

## G. Ablation Studies

Fig. 7(a) shows the highest-entropy channel of $y$ (midpoint 0.8). MoVie allocates more bits to the boat region, highlighting its text-guided focus on perceptually critical content, whereas LIC-TCM and ELIC exhibit more uniform, less semantic-aware distributions.

Fig. 7(b) provides a qualitative comparison of the compressed latent features $\hat{y}$. MoVie exhibits stronger, more localized activations around the fox, clearly outlining semantic regions such as the face and tail—demonstrating the benefit of text-guided fusion. In contrast, the w/o text-guided version of MoVie shows weaker and less focused responses, while VCT produces diffuse activations where the foreground blends into the background, indicating weaker semantic modeling.

## H. Sample Text Descriptions for Fig. 7

To provide context for the visual results shown in Fig. 7, we include the corresponding automatically generated descriptions for each video segment:

**(a) UVG – Bosphorus**
*The video features a serene scene of a dark-colored motorboat with a canopy, cruising on calm waters. The boat is adorned with a red flag on its stern. In the background, a large suspension bridge spans across the water, connecting two land masses covered in lush greenery and dotted with buildings. The sky is clear, suggesting a sunny day, and the water reflects the light, creating a tranquil atmosphere. Other boats are visible in the distance, adding to the sense of a bustling yet peaceful waterway.*

**(b) MCL-JCV – videoSRC20**
*The video features an animated orange fox with a white belly and a blue bib, sitting in a hole in the ground. The fox is holding a fork and knife, and appears to be looking around curiously. In the background, there is a mailbox labeled "MR FOX" and some greenery, including rocks and flowers. The fox then jumps out of the hole and onto the mailbox, causing it to wobble. The fox continues to jump around energetically, holding the utensils, and eventually runs off-screen. The scene transitions to the fox running through a forested area, still holding the fork and knife, and looking excited. The background includes trees, rocks, and patches of grass.*

## I. Caption Templates for Controlled Evaluation

To evaluate the robustness of our method under standardized semantic guidance, we design a manually constructed caption template to simulate typical video descriptions in the absence of human annotations:
*A broad, generic video scene showing one or more primary subjects (people, vehicles, or objects) moving through a typical indoor or outdoor environment under natural or artificial lighting, captured as a wide-to-medium shot with gentle handheld or stabilized camera motion (subtle pan/tilt/track), moderate global motion, and occasional local fast motion around the main subject; the composition centers the subject with clear separation from background, maintaining readable on-screen text, faces, logos, and motion boundaries as top visual priorities, while background textures and repetitive patterns are*

*less critical; colors are neutral to slightly warm with balanced contrast, shadows are soft to moderate, and depth of field is shallow-to-medium to keep the subject prominent; the scene may include brief occlusions, partial reflections, or specular highlights, with mild motion blur on fast regions acceptable; overall pacing is steady, with a small number of salient moments (e.g., a gesture, a directional change) that should remain crisp and legible for downstream perception tasks.*

## J. Details on Large Language Model Usage

In this work, a Large Language Model (LLM) was used only as an auxiliary tool to polish writing, improve grammar, and refine expressions for clarity. It was not used to generate scientific content, design experiments, analyze results, or write substantive parts of the paper. All conceptual ideas, technical contributions, and experiment analyses were conducted entirely by the authors, with the LLM serving solely as a language aid rather than a content creator.

## K. Video-Level Perceptual Metrics

Since FID is an image-based metric and does not explicitly capture temporal consistency, we additionally report FVD (Fréchet Video Distance) and CD-FVD (Ge et al., 2024) as video-specific perceptual metrics.

On UVG with HM as anchor, MoVie achieves the best FVD-based BD-Rate among all compared methods:

| Method | FVD-BD-Rate (%)↓ |
|---|---|
| VTM | −13.03 |
| DCVC-FM | −26.00 |
| **MoVie (Ours)** | **−35.50** |

*Table 9.* FVD-based BD-Rate (%) on UVG with HM as anchor.

We also report representative CD-FVD results at similar bitrates:

| Method | bpp | CD-FVD↓ |
|---|---|---|
| **MoVie (Ours)** | 0.0106 | **321.971** |
| VTM | 0.0116 | 323.170 |
| DCVC-FM | 0.0102 | 333.741 |

*Table 10.* CD-FVD comparison at similar bitrates on UVG.

MoVie achieves the lowest CD-FVD among competitive methods at similar compression ratios, confirming that its perceptual advantage extends beyond frame-level metrics to video-level temporal quality.

## L. Comparison with DCVC-RT

We additionally compare with DCVC-RT (Jia et al., 2025), a recent strong baseline targeting real-time neural video compression. Since the DCVC series does not publicly release training code, we evaluate its official released YUV420 results on UVG and report BD-Rate with HM as anchor:

| Method | FID | LPIPS | PSNR | MS-SSIM |
|---|---|---|---|---|
| MoVie (Ours) | **−45.09** | **−13.03** | 7.62 | −13.98 |
| DCVC-RT | 14.01 | 29.05 | **−25.46** | **−26.91** |

*Table 11.* BD-Rate (%) comparison with DCVC-RT on UVG (HM anchor). DCVC-RT uses official YUV420 results.

The results show a clear trade-off: DCVC-RT is stronger on distortion metrics (PSNR, MS-SSIM), while MoVie achieves substantially better perceptual quality (FID, LPIPS), consistent with our design goal of semantics-aware compression.

## M. Comparison with VideoVAE+

VideoVAE+ (Xing et al., 2025) is primarily designed as a video autoencoding/generation model rather than a low-bitrate entropy-coded video compression method. For a fair reconstruction-level comparison, we report an approximate entropy-based latent-rate proxy for VideoVAE+ (since it does not produce an entropy-coded bitstream), computed as $\text{bpp}_{\text{proxy}} = H(\tilde{z}) \cdot |\tilde{z}|/(B \cdot T \cdot H \cdot W)$, where $\tilde{z}$ is the 8-bit quantized latent tensor and $H(\tilde{z})$ is its empirical Shannon entropy.

| Method | Latent | bpp | PSNR↑ | MS-SSIM↑ | FID↓ | FVD↓ | LPIPS↓ |
|---|---|---|---|---|---|---|---|
| MoVie | – | 0.076 | 37.14 | 0.9698 | 1.398 | 1.840 | 0.2077 |
| MoVie | – | 0.101 | 37.56 | 0.9726 | 1.110 | 1.281 | 0.1872 |
| VideoVAE+ | 4z | 0.456 | 32.78 | 0.9482 | 3.823 | 11.427 | 0.2202 |
| VideoVAE+ | 16z | 1.824 | 34.61 | 0.9690 | 1.494 | 7.750 | 0.1825 |

*Table 12.* Reconstruction comparison with VideoVAE+ on UVG. VideoVAE+ bpp is an entropy-based proxy (not a true codec bitrate). MoVie achieves stronger performance in the low-bitrate regime.

These results show that VideoVAE+ can achieve competitive reconstruction quality with a sufficiently large latent bottleneck, but operates at a much higher effective rate. Our method shows substantially stronger performance in the low-bitrate compression regime.

## N. Visualization results of HEVC

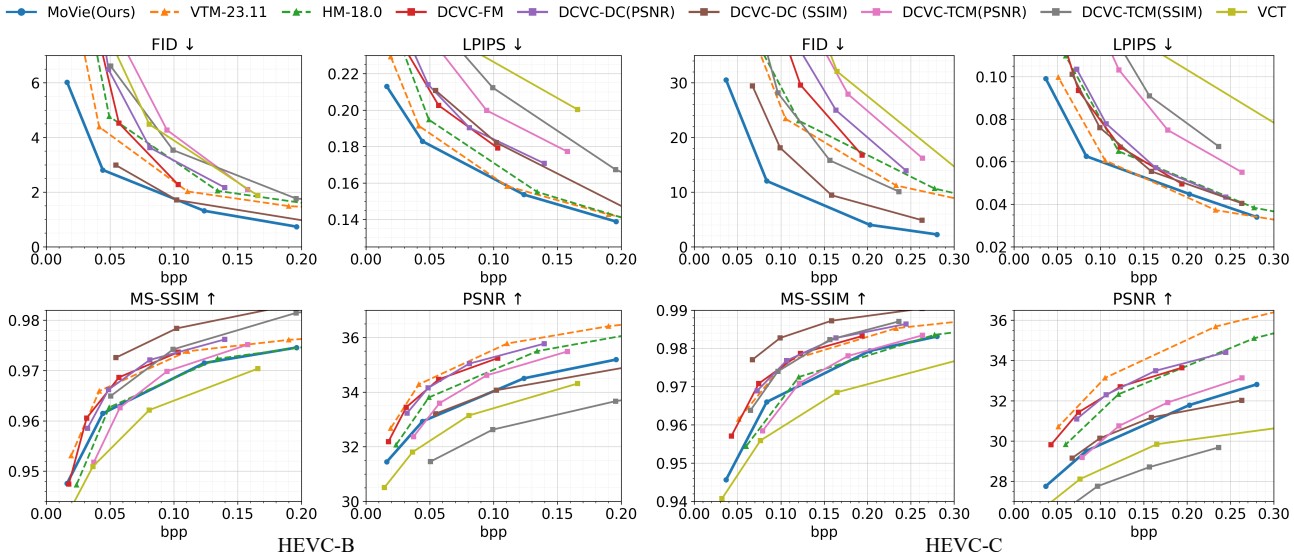

*Figure 8.* Overall compression performance with RD curves (↓ lower is better, ↑ higher is better).

| Method | MAC | HEVC-B | | | | HEVC-C | | | |
|---|---|---|---|---|---|---|---|---|---|
| | | FID | LPIPS | MS-SSIM | PSNR | FID | LPIPS | MS-SSIM | PSNR |
| HM | / | 0 | 0 | 0 | 0 | 0 | 0 | 0 | 0 |
| VTM-23.11 | / | -23.07 | -14.83 | -34.08 | **-32.13** | -9.86 | -20.86 | -31.38 | **-30.33** |
| DCVC-FM | 2051.8 | 15.84 | 29.80 | -27.66 | -21.88 | 35.43 | 4.88 | -32.04 | -16.23 |
| DCVC-DC (PSNR) | 2260.8 | 33.13 | 44.20 | -29.81 | -17.19 | 50.74 | 13.56 | -32.98 | -7.53 |
| DCVC-DC (SSIM) | 2260.8 | -41.48 | 30.73 | **-67.51** | 95.43 | -39.55 | 3.40 | **-61.22** | 77.68 |
| DCVC-TCM (PSNR) | 2322.1 | 75.69 | 115.06 | 11.16 | 28.00 | 85.23 | 84.58 | 8.98 | 62.45 |
| DCVC-TCM (SSIM) | 2322.1 | 37.07 | 117.46 | -37.07 | NaN | -2.49 | 110.31 | -29.56 | NaN |
| VCT | 3115.9 | 41.06 | 239.06 | 57.41 | 122.59 | 78.81 | 205.07 | 57.96 | 232.82 |
| **MoVie (Ours)** | **1144.1** | **-54.35** | **-28.69** | -7.61 | 48.62 | **-64.17** | **-32.66** | -2.72 | 86.00 |

*Table 13.* Comparison on HEVC-B and HEVC-C in terms of BD-Rate (%) using HM-18.0 as the common anchor. BD-Rate is computed as $-M$ for lower-is-better metrics. MAC is measured in kMACs/pixel. NaN indicates failure in BD-rate computation due to excessive performance gap with the anchor.

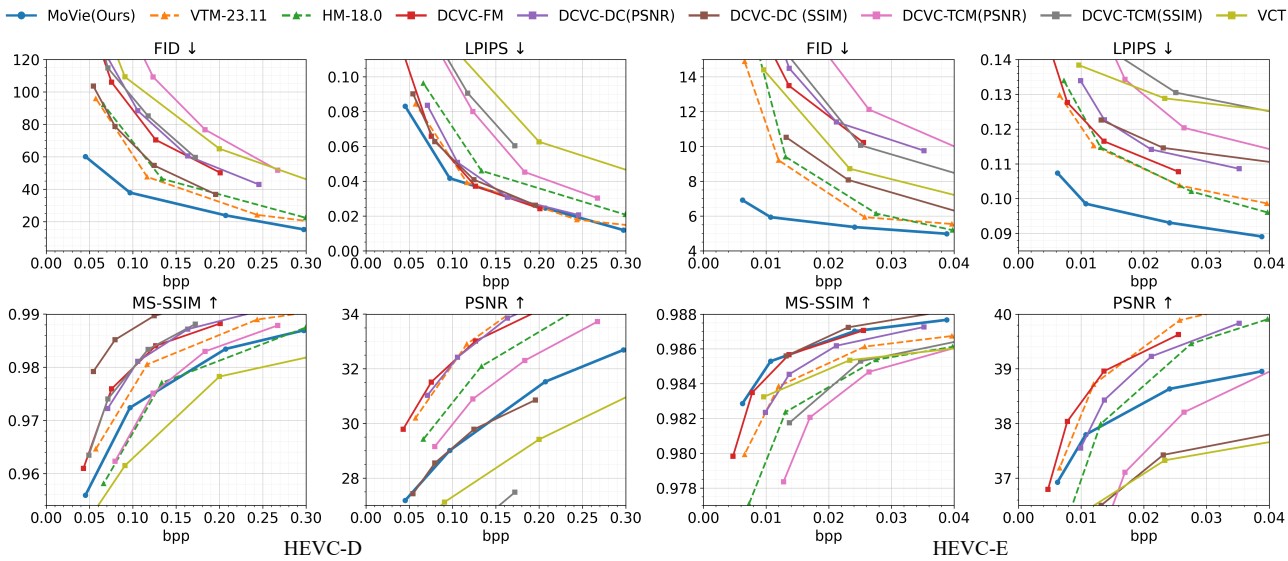

*Figure 9.* Overall compression performance with RD curves (↓ lower is better, ↑ higher is better).

| Method | MAC | HEVC-D | | | | HEVC-E | | | |
|---|---|---|---|---|---|---|---|---|---|
| | | FID | LPIPS | MS-SSIM | PSNR | FID | LPIPS | MS-SSIM | PSNR |
| HM | / | 0 | 0 | 0 | 0 | 0 | 0 | 0 | 0 |
| VTM-23.11 | / | -10.25 | -25.56 | -29.53 | -28.87 | -11.75 | 0.27 | -32.42 | -33.45 |
| DCVC-FM | 2051.8 | 51.34 | -20.29 | -39.69 | -30.91 | 53.64 | 5.26 | -53.42 | -38.44 |
| DCVC-DC (PSNR) | 2260.8 | 63.80 | -12.01 | -39.19 | -24.57 | 82.33 | 48.50 | -37.97 | -11.30 |
| DCVC-DC (SSIM) | 2260.8 | 9.26 | -17.70 | **-66.70** | 80.26 | 23.13 | 116.41 | **-62.41** | 213.39 |
| DCVC-TCM (PSNR) | 2322.1 | 136.33 | 49.48 | 0.72 | 28.50 | 164.01 | 154.28 | 32.91 | 82.85 |
| DCVC-TCM (SSIM) | 2322.1 | 72.79 | 76.59 | -38.85 | NaN | 90.99 | 293.92 | -3.74 | NaN |
| VCT | 3115.9 | 103.36 | 104.01 | 39.25 | 193.70 | 48.52 | 277.70 | -18.25 | 133.63 |
| **MoVie (Ours)** | **1144.1** | **-41.46** | **-34.04** | -9.27 | 77.72 | **-58.44** | **-63.09** | -60.31 | -0.62 |

*Table 14.* Comparison on HEVC-D and HEVC-E in terms of BD-Rate (%) using HM-18.0 as the common anchor. BD-Rate is computed as $-M$ for lower-is-better metrics. MAC is measured in kMACs/pixel. NaN indicates failure in BD-rate computation due to excessive performance gap with the anchor.

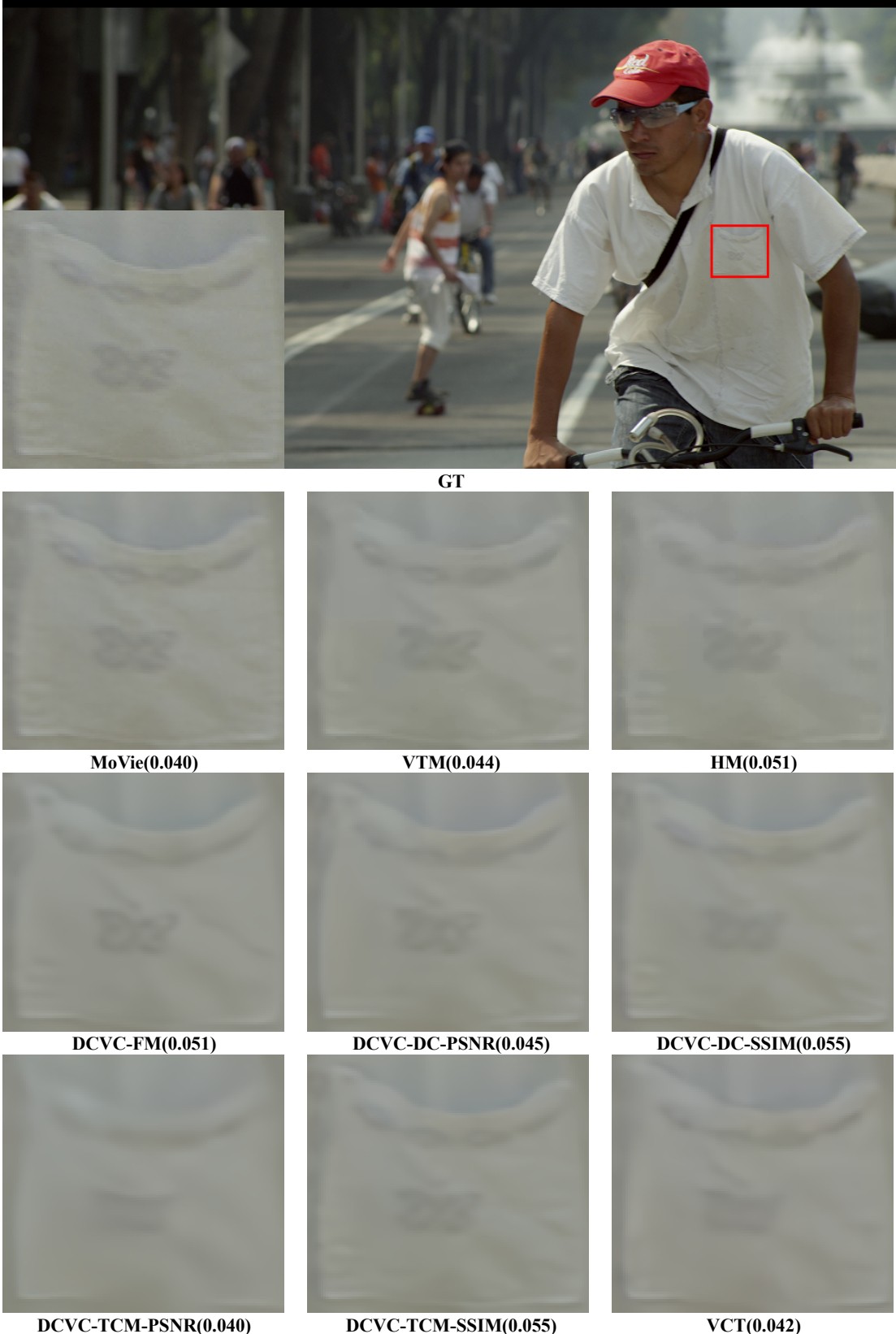

*Figure 10.* Subjective quality comparison on the MCL-JCV *videoSRC02* sequence.

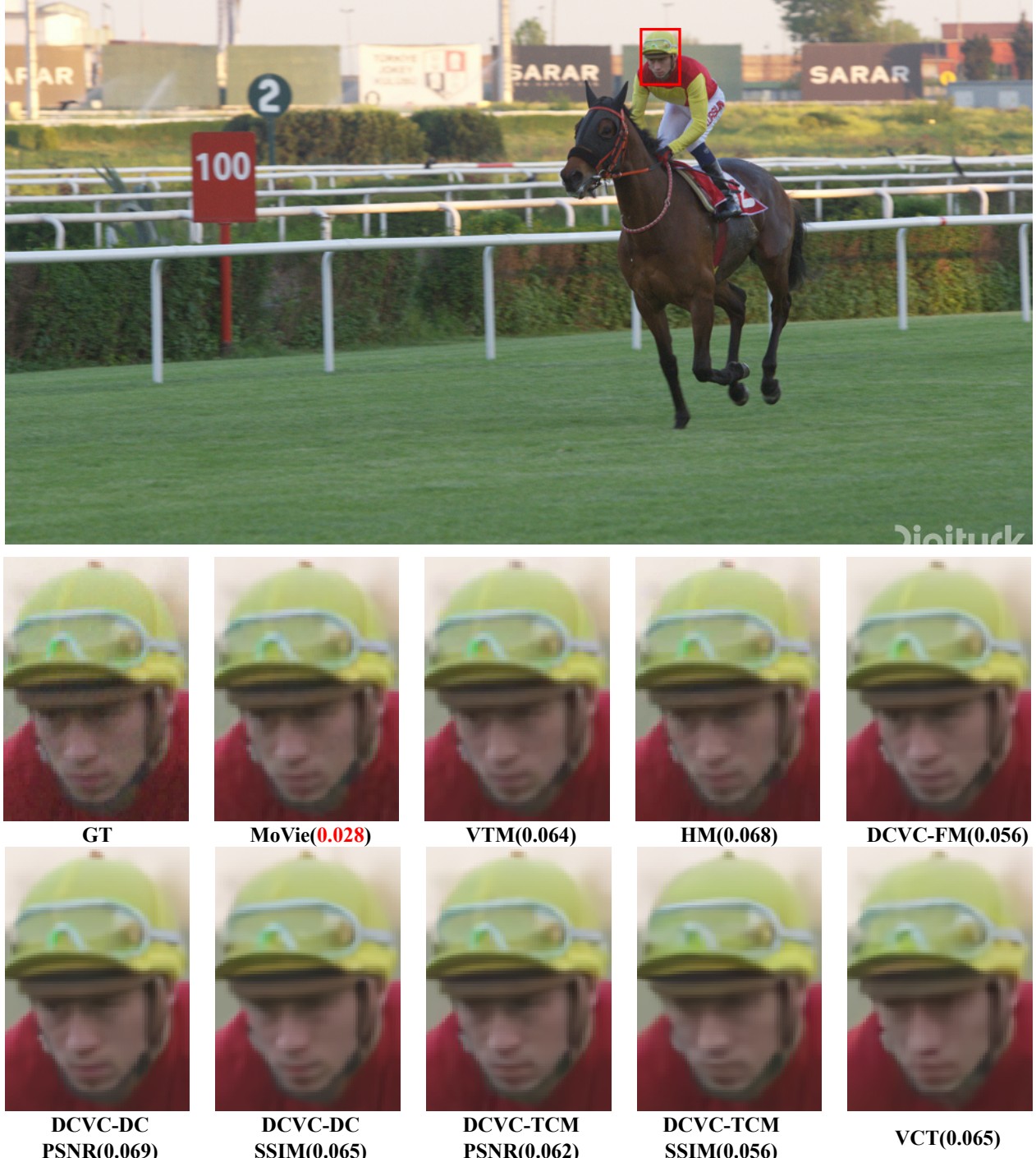

*Figure 11.* Subjective quality comparison on the UVG *Jockey* sequence.

