# OpenReview forum: "MoVie: Multimodal Video Compression with Text Guidance"
_ICML.cc/2026/Conference — ICML 2026 regular_

### Official Review · Reviewer_Nt8V · 2026-03-07

**Soundness:** 1
**Presentation:** 3
**Significance:** 3
**Originality:** 2
**Overall Recommendation:** 4
**Confidence:** 4

**Summary:**

The paper introduces MoVie, a learned video compression approach that incorporates textual semantics to guide compression decisions. The main idea is to use text descriptions of video content to bias the encoder toward preserving semantically important information, aiming to improve perceptual quality at low bitrates.

Text information is injected through a two-stage process. An extractor uses cross-attention between text embeddings and video features to produce semantic signals, and an injector merges these signals into the video latent representation.

Training uses a combination of rate–distortion, perceptual, and cross-modal alignment losses. Experiments on UVG and MCL-JCV evaluate perceptual and distortion metrics such as FID, LPIPS, PSNR, and MS-SSIM. Results indicate improved perceptual quality compared to some learned compression baselines while maintaining lower computational cost than some Transformer-based approaches.

**Compliance With Llm Reviewing Policy:**

Affirmed.

**Final Justification:**

The authors’ rebuttal effectively addresses my main concerns. The human perceptual study validates the semantic guidance claim, the ablations clarify component contributions, and the DCVC‑RT evaluation strengthens baseline comparisons.

**Key Questions For Authors:**

1. The central claim is that text guidance helps preserve semantically important regions. Have you considered or conducted a human perceptual study to validate this? If the authors can provide evidence that semantic regions are indeed better preserved, it would strengthen the main contribution significantly.

2. The evaluation does not include recent strong baselines like DCVC‑RT. Could the authors clarify whether retraining such baselines with the same loss combination was attempted or feasible?

3. Multiple architectural changes are introduced simultaneously. Can the authors provide ablation studies isolating the contribution of each component (Transformer–CNN backbone, extractor/injector, entropy modifications)?

4. How does the additional computational overhead from text guidance and cross-modal attention affect inference speed and deployability compared to standard learned compression pipelines?

**Limitations:**

yes

**Strengths And Weaknesses:**

## Strengths

- Proposes a technically sound multimodal video compression framework that integrates text-derived semantic guidance into a learned compression pipeline.
- Builds on established components, including Transformer-based video encoders, 3D CNN processing, and entropy models.
- Training combines rate, distortion, and perceptual losses, following standard practice in learned perceptual compression.
- Evaluated on standard datasets (UVG and MCL-JCV) across multiple rate-distortion operating points.

## Weaknesses

- Central claim of preserving semantically important regions is not directly tested. Evaluation relies solely on automated metrics (LPIPS, FID), which are insufficient proxies for human perception. No human perceptual study is provided to validate semantic guidance benefits.
- Comparisons to prior work are limited; the paper omits strong recent baselines such as DCVC‑RT [1]. Even including them would require retraining with the same loss combination, since perceptual quality depends heavily on the training objective.
- The method introduces multiple components (hybrid Transformer–CNN backbone, text-guided extractor/injector, modified entropy model) without clear ablations to isolate their individual contributions.
- Overall contribution is primarily engineering-driven, with inadequate evaluation to support the main claims.

[1] Jia, Zhaoyang, et al. "Towards practical real-time neural video compression." Proceedings of the Computer Vision and Pattern Recognition Conference. 2025.

---

> ### Author Rebuttal · Authors · 2026-03-31
>
> ## Response
>
> We thank the reviewer for the detailed and valuable suggestions.
>
> ### 1. On lack of direct validation of semantic region preservation
> We agree that LPIPS and FID alone are insufficient proxies for human perception. To address this, we conducted an additional human perceptual study.
>
> **Study Design.**
>
> We recruited 20 naive subjects (non-experts in video compression) and selected 20 videos from four datasets and resolutions (7 UVG, 5 MCL-JCV, 4 HEVC-C, and 4 HEVC-D). Each subject evaluated all 20 videos (within-subject design), resulting in 400 trials per baseline comparison. Evaluation was conducted at low bitrates (bpp ≈ 0.01), where semantic degradation is more noticeable. In each trial, subjects viewed the reference video together with our result and a baseline, and chose the one perceptually closer to the reference.
>
> **Results.**
>
> |Baseline|Win Rate|95% CI|BT Score|
> |---|---|---|---|
> |DCVC-FM|73.75%|[69.1%, 77.9%]|-1.033|
> |VTM|78.00%|[73.7%, 81.8%]|-1.266|
> |PLVC|85.50%|[81.7%, 88.6%]|-1.774|
>
> Our method is preferred in 73.75%, 78.00%, and 85.50% of comparisons against DCVC-FM, VTM, and PLVC, respectively. A Bradley–Terry model also assigns all baselines negative scores, confirming consistent subjective preference for our method.
>
> Although not pixel-level annotation, this study provides direct human-subjective evidence beyond LPIPS/FID that semantic guidance improves perceptual quality and better preserves semantically salient information.
>
> ### 2. On missing strong baselines such as DCVC-RT
> We thank the reviewer for this suggestion. To address this concern, we additionally evaluated DCVC-RT and report the results below.
>
> We did not retrain DCVC-RT with our loss combination because the DCVC series does not publicly release its training code. Instead, we followed its official released models and evaluation protocol. As stated in the repository, *“We highly suggest testing YUV420 content. To test RGB content, please refer to the DCVC-FM folder,”* so we evaluated its released YUV420 compressed results at four rate points and compared them with MoVie under the HM anchor on UVG.
>
> |Method|FID|LPIPS|PSNR|MS-SSIM|
> |---|---:|---:|---:|---:|
> |MoVie(Ours)|-45.09|-13.03|7.62|-13.98|
> |DCVC-RT|14.01|29.05|-25.46|-26.91|
>
> These results show a clear trade-off: DCVC-RT is stronger on distortion metrics, while MoVie is better on the perceptual metrics targeted here. We also include PLVC to partially account for training-objective differences.
>
> ### 3. On missing component-wise ablations
> Thank you for this valuable comment. We agree that the original manuscript did not sufficiently isolate the contributions of individual components. We therefore conducted a clearer progressive ablation.
>
> Starting from the baseline, we progressively add the S-C entropy model, VideoTCM, and the text-guided module, and evaluate their impact using BD-Rate under LPIPS, FID, and PSNR:
> |Ablation Setting|BD-Rate(LPIPS)|BD-Rate(FID)|BD-Rate(PSNR)|
> |---|---:|---:|---:|
> |Baseline(anchor)|0.00%|0.00%|0.00%|
> |+S-C Entropy Model|-45.39%|-52.04%|-5.70%|
> |+S-C Entropy Model+VideoTCM|-73.60%|-71.38%|-19.51%|
> |+S-C Entropy Model+VideoTCM+Text-Guided|-87.07%|-81.94%|-16.07%|
>
> The results show complementary gains from all components. The S-C entropy model improves all three metrics, VideoTCM gives the best PSNR BD-Rate, and the text-guided module gives the best perceptual performance, reaching -87.07% in LPIPS and -81.94% in FID.
>
> ### 4. On the overall contribution being engineering-driven
> We respectfully disagree with this characterization. The central contribution is not a combination of existing modules, but a new compression paradigm: using natural language as a semantic prior to guide rate allocation and perceptual reconstruction in a video codec. Prior learned codecs optimize rate–distortion objectives over pixel or latent statistics alone; our work demonstrates that freely available text descriptions can serve as a structured, human-aligned prior that redirects coding resources toward semantically important regions. This formulation—bridging cross-modal language understanding and video compression—is itself a novel problem setting, independent of the specific modules used to instantiate it.
>
> ### 5. On computational overhead and deployability
> Thank you for this important question. We agree that text guidance introduces additional overhead compared with standard learned compression pipelines. The main extra cost comes from text extraction rather than the cross-modal module itself.
>
> As detailed in our response to Reviewer fUKV on the LLaVA captioning cost, under the same 96-frame setting, the runtime of the semantic-guided codec is 324 ms. Including LLaVA-Video-7B-Qwen2 text generation (132 ms per 96-frame clip), the total encoding time becomes 456 ms, still comparable to DCVC-DC (458 ms) and DCVC-FM (439 ms), and lower than VCT (614 ms). The overhead is encoder-side only; when text metadata is available, no extra inference cost is incurred.

---

> > ### Author Rebuttal · Reviewer_Nt8V · 2026-04-02
> >
> > Thank you for the detailed clarifications and the additional experiments. The inclusion of a human perceptual study directly addressing semantic region preservation is particularly valuable, as it strengthens the core claim of the paper. The progressive ablations and the evaluation against DCVC‑RT also help address my previous concerns about component contributions and baseline comparisons.
> >
> > Given these clarifications, I recommend that the human study results be incorporated into the main paper, as they provide crucial evidence supporting the benefits of text-guided compression. With these additions, my main concerns are largely addressed.
> >
> > Accordingly, I am updating my score.

---

> > > ### Author Response · Authors · 2026-04-07
> > >
> > > Thank you very much for the careful follow-up and for updating your score. We are glad that the additional experiments and clarifications helped address your concerns. We also appreciate your suggestion regarding the human perceptual study. If the paper is accepted, we will incorporate these results into the main paper, as they provide important evidence supporting the benefits of text-guided compression.

---

### Official Review · Reviewer_fUKV · 2026-03-11

**Soundness:** 3
**Presentation:** 3
**Significance:** 2
**Originality:** 4
**Overall Recommendation:** 3
**Confidence:** 4

**Summary:**

The paper implements a method for text-guided video compression. By leveraging a large language model to extract video content descriptions and injecting them into the encoder to assist with semantic feature extraction, the proposed approach achieves superior performance on perceptual metrics like LPIPS.

**Compliance With Llm Reviewing Policy:**

Affirmed.

**Key Questions For Authors:**

See weaknesses.

**Limitations:**

See weaknesses.

**Strengths And Weaknesses:**

Strengths
1. The paper successfully integrates text into the video compression framework to actively assist and enhance semantic feature extraction。
2. It effectively utilizes the power of large multimodal models (LLaVA, CLIP) to automatically generate and extract high-level semantic priors.
3. The authors design a powerful fusion module (Text-VideoTCM) that combines 3D CNNs and Video Swin Transformers to seamlessly integrate text information with spatiotemporal video features

Weaknesses
1. The reported encoding complexity and inference times (Table 3) seem to omit the significant time and computational cost required by the large language model (LLaVA) to extract the text descriptions. This omission makes the efficiency comparisons with traditional or non-LLM baselines incomplete and potentially unfair.
2. Although the abstract and introduction explicitly claim that the proposed method improves "temporal coherence," the paper lacks any quantitative metrics (such as warping error or temporal loss) or dedicated experiments to actually measure and prove this frame-to-frame stability.

---

> ### Author Rebuttal · Authors · 2026-03-31
>
> ## Response
> We thank the reviewer for the insightful comments.
>
> ### 1. On omission of LLaVA captioning cost
> Thank you for this important comment. We agree that omitting the cost of LLaVA-based text extraction makes the efficiency comparison incomplete, and this should be clarified more explicitly in the paper.
>
> Specifically, the encoding time originally reported in Table 3 measures only the runtime of the semantic-guided codec itself, and does not include the additional inference time of LLaVA for text generation. We acknowledge that comparing this number directly with conventional baselines is not fully fair. In the revision, we will explicitly separate the codec runtime and the text-extraction overhead, and report the total encoding time for a more complete comparison.
>
> To quantify this cost, we additionally measured the latency of LLaVA-Video-7B-Qwen2 for text generation under the same 96-frame setting. The average text-generation overhead is 132 ms per frame. After including this cost, the total encoding time of our method becomes 456 ms, which is still comparable to DCVC-DC (458 ms), DCVC-FM (439 ms), and notably lower than VCT (614 ms).
>
> |Method|Enc.time|Text.time|Total Enc.time|
> |:--|--:|--:|--:|
> |DCVC-DC|458 ms|–|458 ms|
> |DCVC-FM|439 ms|–|439 ms|
> |VCT|614 ms|–|614 ms|
> |MoVie|324 ms|132 ms|456 ms|
>
> At the same time, we would like to emphasize that our framework does not require text descriptions to be extracted online by an MLLM in every practical deployment. In many real-world scenarios, videos already come with associated text metadata, such as titles or uploader-provided descriptions, which can be directly used as semantic guidance at essentially no additional inference cost.
>
> To verify this practical alternative, we compared two text sources on ten online videos under the same 96-frame setting: (1) LLaVA-generated text and (2) original YouTube text. While generated text achieves better overall performance, pre-existing text still provides competitive semantic guidance and yields reasonable perceptual quality.
>
> | Text Source | bpp | PSNR↑ | MS-SSIM↑ | FID↓ | LPIPS↓ |
> |:--|--:|--:|--:|--:|--:|
> | Generated Text | 0.167 | 40.469 | 0.994 | 5.975 | 0.103 |
> | Original YouTube Text  | 0.187 | 39.629 | 0.989 | 8.967 | 0.117 |
>
> Therefore, online LLaVA inference should be viewed as one possible usage mode rather than a mandatory component of the pipeline. In the revised manuscript, we will clarify that the original encoding time excluded LLaVA inference, add the full latency breakdown, and discuss the practical setting where pre-existing text can be used directly for fairer and more transparent efficiency comparisons.
>
> ### 2. On the claim of improved temporal coherence
>
> We thank the reviewer for this valid and important comment. We agree that the original manuscript lacked a dedicated quantitative evaluation to directly support the temporal coherence claim, and we therefore conducted additional experiments on the **96-frame** UVG setting.
>
> **Evaluation Protocol.**
> We estimate optical flow between adjacent reconstructed frames using pretrained RAFT [Teed et al., ECCV 2020], warp the reconstructed frame at time t toward t+1, and compute the mean absolute pixel error against the reconstructed frame at t+1 (Mean Warping Error, ↓). A lower value indicates better frame-to-frame temporal stability.
>
> **Fair Comparison via BD-Rate.**
> Since different methods operate at different bitrates, a single-point comparison of warping error would be confounded by compression strength. We therefore evaluate each method across four rate points and compute BD-Rate with Mean Warping Error as the quality axis, using HM as the anchor. This allows a fair, rate-controlled comparison of temporal coherence across methods.
> | Method | BD-Rate (Mean Warping Error↓) |
> |--------|----------------------|
> | HM | 0% (anchor) |
> | VTM | −14.42% |
> | DCVC-FM | +3.19% |
> | **MoVie (Ours)** | **−15.88%** |
>
> A negative BD-Rate indicates that the method achieves equivalent temporal coherence at a lower bitrate compared to HM. Our method achieves a BD-Rate of **−15.88%**, outperforming all compared methods including VTM (−14.42%) and DCVC-FM (+3.19%), confirming that semantic guidance from text descriptions helps maintain consistent high-level content across frames and reduces perceptual temporal fluctuations under compression.
>
> We will include these quantitative results and representative visual examples of reduced flickering in the revised manuscript, and revise the wording in the abstract and introduction to precisely reflect the presented evidence.
>
> In summary, the two concerns are now both addressed: full latency breakdown is provided in Table above, and Mean Warping Error BD-Rate directly quantifies temporal coherence.

---

> > ### Author Rebuttal · Reviewer_fUKV · 2026-04-03
> >
> > I thank the authors for the additional results. While the data is now more complete, it remains difficult to determine whether the gains are truly driven by semantic guidance or by other factors. Therefore, I maintain my current assessment.

---

> > > ### Author Response · Authors · 2026-04-07
> > >
> > > ## Response
> > > Thank you for the follow-up. We understand your concern regarding whether the observed gains are truly driven by semantic guidance, rather than by unrelated factors such as increased parameter count or structural bias.
> > >
> > > We would like to clarify that the manuscript already provides several complementary pieces of evidence addressing this question:
> > >
> > > **1. Controlled inference-time ablation (Table 4, Sec. 4.5).**
> > > In this experiment, exactly the same trained model is used, and only the text input is changed at inference time:
> > >
> > > | Text Input | bpp | PSNR↑ | MS-SSIM↑ | FID↓ | LPIPS↓ |
> > > |---|---:|---:|---:|---:|---:|
> > > | Ground Truth | **0.011** | **34.61** | **0.94930** | **6.731** | **0.19388** |
> > > | Rephrased Text | 0.011 | 34.57 | 0.94928 | 6.731 | 0.19411 |
> > > | Irrelevant Text | 0.035 | 29.35 | 0.92912 | 15.129 | 0.21280 |
> > > | Without Text | 0.032 | 29.77 | 0.93068 | 14.694 | 0.21164 |
> > > | Template Text | 0.011 | 34.26 | 0.93934 | 11.657 | 0.20340 |
> > >
> > > Semantically aligned captions (“Ground Truth”) and semantically equivalent paraphrases (“Rephrased Text”) yield nearly identical results, whereas irrelevant or missing text causes substantial degradation (e.g., FID increases from 6.731 to 15.129 / 14.694, and LPIPS increases from 0.19388 to 0.21280 / 0.21164). Since the model itself is unchanged across all settings, this pattern is more consistent with the effect of semantic relevance than with parameter count or architectural bias alone.
> > >
> > > **2. Structural ablation on where text is injected (Table 2, Sec. 4.4).**
> > > The manuscript compares encoder-only, decoder-only, and dual-side text injection. Our encoder-side design (Enc-T / Dec-V) achieves the best perceptual quality, while decoder-only or dual-side injection does not further improve performance and can even hurt reconstruction quality. This suggests that the gain is not due to simply adding more branches or parameters, but to using text in a way that effectively shapes the latent representation at the encoder side.
> > >
> > > In the same section, we also compare alternative text strategies such as Global Text, Copy Text, and Constant Text. These results show that frame-agnostic or non-informative text is ineffective, while the proposed two-stage design performs best. This further supports that the improvement depends on informative, content-dependent semantic guidance, rather than on the mere presence of an auxiliary text modality.
> > >
> > > **3. Qualitative evidence on encoder response (Fig. 6, Sec. 4.5).**
> > > This interpretation is also consistent with the qualitative evidence in Fig. 6. The visualization of the top-8 highest-entropy channels shows that, when the text correctly describes both birds, the encoder response focuses on both semantic targets; with irrelevant or no text, the response becomes more diffuse and less target-aware. Template text provides partial guidance, but remains clearly weaker than semantically aligned text. This qualitative behavior is aligned with the quantitative trends in Table 4.
> > >
> > > Taken together, these results support the same conclusion from multiple angles: the observed gains are associated with semantically aligned, informative text guidance, rather than with unrelated model changes alone.
> > >
> > > We realize that this point was not highlighted clearly enough in our previous response, and we hope this clarification better addresses the reviewer’s concern.

---

### Official Review · Reviewer_uqiF · 2026-03-12

**Soundness:** 3
**Presentation:** 2
**Significance:** 3
**Originality:** 2
**Overall Recommendation:** 4
**Confidence:** 3

**Summary:**

MoVie is a multimodal video compression framework that uses text guidance to improve low-bitrate video reconstruction, especially in terms of perceptual quality. It builds a video-specific encoder-decoder with a hybrid Video Swin Transformer and 3D CNN backbone, rather than relying on a purely image-based design. The key idea is to align a clip-level text description with individual video frames through a two-stage text fusion module, so the model can inject frame-relevant semantic information during encoding. It also introduces a history-conditioned entropy model to better exploit temporal, spatial, and channel dependencies for more efficient compression.

**Compliance With Llm Reviewing Policy:**

Affirmed.

**Final Justification:**

The author's feedback has addressed most of my concerns, and I'd like to maintain my original rating.

**Key Questions For Authors:**

- The improvements over TACO are not fully addressed. Although this work extend text guidance from image to video compression, how the proposed modules leverage texts for temporal modeling specifically? It also helps to discuss the differences between VideoVAE+ etc.

**Limitations:**

This paper didn't discuss its limitations.

**Strengths And Weaknesses:**

Strengths:

- The proposed leverages text prior in video compression and achieves SOTA performance. Figure 6 also demonstrates semantic decomposition and alignment between words and video content.

Weaknesses:

- FID instead of FVD is used to measure video reconstruction quality. More advanced metrics (e.g. CD-FVD) are also encouraged.

- Ablation study only investigates the impact of text quality in a well-trained model. However, to validate the contributions of each design components, it's expected to train multiple models with different text input / without certain modules to compare.

- There are also more text-aware video autoencoders, such as VideoVAE+[2]. Although they primarily serve video generation, which is a more complex task, their compression/reconstruction performance can still be compared.

- Visual result presentation is limited. Only several static frame comparisons, without video files in the supplementary zip.

[1] On the Content Bias in Fréchet Video Distance (Ge et al. CVPR 2024)
[2] VideoVAE+: Large Motion Video Autoencoding with Cross-modal Video VAE (Xing et al. ICCV 2025)

---

> ### Author Rebuttal · Authors · 2026-03-31
>
> ## Response
> We thank the reviewer for the helpful comments and suggestions.
>
> ### 1. On FID vs. FVD / CD-FVD
> Thank you for this valuable suggestion. We agree that FID alone is not sufficient for evaluating video reconstruction quality, since it is an image-based perceptual metric and does not explicitly capture temporal consistency. We therefore additionally report FVD as a video-specific perceptual metric.
>
> On UVG (HM-18.0 as anchor), MoVie achieves the best FVD-based BD-Rate among all valid compared methods:
> |Method|FVD-BD-Rate(%)↓|
> |---|---:|
> |VTM|-13.03|
> |DCVC-FM|-26.00|
> |MoVie(Ours)|-35.50|
>
> These results show that the perceptual gains of MoVie remain valid under a video-level metric. We also appreciate the suggestion of CD-FVD and additionally report representative CD-FVD results at similar bitrates on UVG:
> |Method|bpp|CD-FVD↓|
> |---|---:|---:|
> |MoVie (Ours)|0.0106|321.971|
> |VTM|0.0116| 323.170|
> |DCVC-FM|0.0102|333.741|
>
> MoVie achieves the lowest CD-FVD among these competitive methods at similar compression ratios, which further supports that our method provides better perceptual video reconstruction quality beyond FID alone.
>
> ### 2. On ablation design
> Thank you for this valuable comment. We agree that validating the contribution of each design component requires training multiple variants with different modules enabled or removed. To address this concern, we conducted a clearer progressive ablation by retraining models with different component combinations.
>
> Starting from the baseline, we progressively add the S-C entropy model, VideoTCM, and the text-guided module, and evaluate their impact using BD-Rate under LPIPS, FID, and PSNR:
> |Ablation Setting|BD-Rate(LPIPS)|BD-Rate(FID)|BD-Rate(PSNR)|
> |---|---:|---:|---:|
> |Baseline(anchor)|0.00%|0.00%|0.00%|
> |+S-C Entropy Model|-45.39%|-52.04%|-5.70%|
> |+S-C Entropy Model+VideoTCM|-73.60%|-71.38%|-19.51%|
> |+S-C Entropy Model+VideoTCM+Text-Guided|-87.07%|-81.94%|-16.07%|
>
> These results show complementary gains from all components. The S-C entropy model improves all three metrics, VideoTCM gives the best PSNR BD-Rate, and the text-guided module provides the strongest perceptual gains, reaching -87.07% in LPIPS and -81.94% in FID. Thus, the ablation goes beyond text quality in a fixed model and explicitly compares retrained variants with and without major modules.
>
> ### 3. On comparison with VideoVAE+ and related models
> Thank you for this suggestion. We agree that VideoVAE+ and similar text-aware video autoencoders are relevant related works. However, they are mainly designed for video generation/tokenization rather than video compression under a rate–distortion or rate–perception objective, so a direct comparison is only meaningful under a matched reconstruction setting. We also agree that using such models as video autoencoders may be a promising future direction for video compression, although practical deployment still requires solving issues such as entropy coding and bitrate control. We will cite these works more explicitly in the final manuscript and clarify their differences from our setting as well as their future potential.
>
> ### 4. On limited visual presentation
> We agree that static frame comparisons alone are insufficient to fully demonstrate temporal perceptual quality, and that reconstructed videos would better show temporal consistency, flickering, and other perceptual differences. We also acknowledge that the current supplementary zip does not include video files. As the rebuttal stage only allows an author response rather than additional supplementary uploads, we are unable to provide videos here.
>
> To partially compensate for this limitation, we additionally report a quantitative temporal-consistency metric, Mean Warping Error, which provides a more direct assessment of frame-to-frame stability. Its evaluation protocol is described in our response to Reviewer fUKV, Comment 2.
> | Method | BD-Rate (Mean Warping Error↓) |
> |---|---:|
> | HM | 0% (anchor) |
> | VTM | -14.42% |
> | DCVC-FM | +3.19% |
> | MoVie (Ours) | -15.88% |
>
> These results show that MoVie achieves the best Mean Warping Error-based BD-Rate, indicating that its advantage extends beyond static-frame quality to temporal consistency. We nevertheless agree that reconstructed videos would further strengthen the presentation.
>
> ### 5. On the difference from prior text-guided methods such as TACO
> We appreciate this point. Our work differs from prior text-guided image compression methods such as TACO in that text is integrated into a temporally-aware video codec rather than used as frame-level guidance alone. Specifically, the extractor/injector design operates on video features that already encode temporal context, allowing semantic cues to influence motion-aware and context-aware compression decisions across frames. Therefore, the role of text in our method is not only to enhance frame-level semantics, but also to interact with temporally modeled video representations in the video compression pipeline.

---

> > ### Author Rebuttal · Reviewer_uqiF · 2026-04-01
> >
> > I appreciate the authors' feedback. It has addressed part of my concerns, including the FVD metrics and ablation experiments. Meanwhile, I might want to further discuss a few points:
> >
> >  - Reviewer RoSU mentions the proposed model relies on CLIP performance and generalization. Since the proposed method is for videos, video-version of language-vision contrastive learning, such as VideoCLIP, X-CLIP and InternVideo should be considered.
> >
> > - For VideoVAE+ etc., being designed for video generation would be a burden for these models as they need KL bottleneck on their latent, which harms the expressiveness and reconstruction. Therefore, I'd still encourage a comparison.
> >
> > - The official rebuttal instructions allow anonymous links for external figures and images/videos (please double check the email and FAQ). So please add sufficient and comprehensive visualizations for evaluation and comparison.

---

> > > ### Author Response · Authors · 2026-04-07
> > >
> > > ## Response
> > > We thank the reviewer for the constructive follow-up questions and suggestions.
> > >
> > > ### 1. On video-specific vision-language encoders
> > > We agree that this is an important question, especially since our framework is designed for video compression. In our current design, however, the text branch mainly serves as a source of semantic conditioning (scene/content priors), while temporal dynamics are modeled by the video encoder and compression backbone. Therefore, the role of the text encoder in our framework is different from that in video-language models such as VideoCLIP, X-CLIP, and InternVideo, whose main distinction lies in joint video-text representation learning rather than in a standalone text branch.
> > >
> > > As a result, a meaningful comparison with such models would require integrating their full video-text modeling into our framework and retraining the complete system, rather than simply swapping the text encoder. This is unfortunately not feasible within the rebuttal period.
> > >
> > > As indirect evidence, Table 4 in the main paper shows that semantic guidance is beneficial in our framework: rephrased text yields nearly identical results, while removing text guidance or using irrelevant text substantially degrades perceptual quality (e.g., FID rises from 6.731 to 14.694/15.129, and LPIPS increases from 0.19411 to 0.21164/0.21280). This suggests that performance mainly depends on whether the text provides semantically relevant guidance, rather than on its exact wording, and indicates that a standard CLIP text encoder is already effective in our framework.
> > >
> > > ### 2. On comparison with VideoVAE+ style models
> > > We agree that a comparison with VideoVAE+ style models is informative, and we therefore additionally evaluated VideoVAE+ in a reconstruction setting.
> > >
> > > For fairness, we report an approximate entropy-based latent-rate proxy for VideoVAE+, rather than a true compressed bitrate, since VideoVAE+ does not produce an entropy-coded bitstream in our evaluation pipeline. Specifically, we first quantize the continuous latent tensor into 8-bit symbols using min-max normalization (256 bins), then estimate the empirical Shannon entropy of the quantized latent values from their histogram distribution. The resulting proxy is computed as:
> > >
> > > bpp_proxy = H(z_tilde) * |z_tilde| / (B * T * H * W),
> > >
> > > where z_tilde is the 8-bit quantized latent tensor, H(z_tilde) is its empirical Shannon entropy in bits per symbol, |z_tilde| is the total number of latent elements, and B, T, H, W denote the batch size, number of frames, and spatial resolution of the input video. This quantity should be interpreted only as a rough proxy for bottleneck size / representation cost, rather than a true codec bitrate, since it does not include an actual entropy model, bitstream generation, or side-information overhead.
> > >
> > > |Method|Latent Size|bpp|PSNR↑|MS-SSIM↑|FID↓|FVD↓|LPIPS↓|
> > > |---|---|---:|---:|---:|---:|---:|---:|
> > > |MoVie(Ours)|-|0.076|37.1377|0.9698|1.3984|1.8398|0.2077|
> > > |MoVie(Ours)|-|0.101|37.5553|0.9726|1.1098|1.2805|0.1872|
> > > |VideoVAE+|4z|0.456|32.7788|0.9482|3.8227|11.4266|0.2202|
> > > |VideoVAE+|16z|1.824|34.6108|0.9690|1.49351|7.7499|0.1825|
> > >
> > > These results show that VideoVAE+ can achieve competitive reconstruction quality when a sufficiently large latent bottleneck is allowed. However, compared with our method, it operates at a much higher effective bottleneck rate in this evaluation. In contrast, our method shows substantially stronger performance in the low-bitrate compression regime studied in this work. For example, even at 0.076 bpp, our method achieves higher PSNR and markedly better FID/FVD than the 16z VideoVAE+ setting in our evaluation.
> > >
> > > Overall, we view this comparison as informative for reconstruction capability, but not as a fully apples-to-apples compression comparison. It also highlights that VideoVAE+ is primarily designed as a video autoencoding / generation model, rather than a low-bitrate entropy-coded video compression method.
> > >
> > > ### 3. On additional anonymous visualizations
> > > Thank you for this helpful suggestion. To strengthen the qualitative comparison, we provide an anonymous visualization page containing reconstructed videos and side-by-side comparisons for all UVG sequences. All visualizations are conducted at matched bitrates. The average bitrate across all UVG sequences is approximately 0.011 bpp for each method.
> > >
> > > https://anonymous.4open.science/r/icml5110-B70E
> > >
> > > The page includes comparisons with strong baselines, including VTM and DCVC-FM, at closely matched bitrates. For each video, we additionally highlight representative local regions with zoomed-in views to better illustrate perceptual details, temporal consistency, and flickering artifacts. We hope these supplementary visualizations provide a clearer and more comprehensive qualitative evaluation. Across these sequences, our method generally preserves semantically important local details more faithfully and exhibits reduced temporal flickering compared with the baselines.

---

### Official Review · Reviewer_RoSU · 2026-03-13

**Soundness:** 3
**Presentation:** 3
**Significance:** 3
**Originality:** 3
**Overall Recommendation:** 4
**Confidence:** 3

**Summary:**

The authors propose MoVie, a multimodal video compression framework that integrates high-level textual semantics into the compression pipeline to improve perceptual quality. The core contribution is a Text-guided Video Transformer-CNN Mixed block that processes spatial and temporal dynamics in parallel. To fuse modalities, the authors introduce a dual-stage text fusion mechanism using a frozen CLIP text encoder. The proposed method demonstrates strong rate-distortion performance on perceptual metrics (FID, LPIPS) at lower computational costs compared to recent learned baseline codecs.

**Compliance With Llm Reviewing Policy:**

Affirmed.

**Key Questions For Authors:**

1. The evaluation standardly considers the first 96 frames of each sequence. Given the history-conditioned entropy model and the generative nature of the text injection, how does the model handle long-term temporal error propagation and the potential for flickering or semantic hallucination over sequences longer than 300 frames?
2. The pipeline relies heavily on the frozen CLIP text encoder. How does the codec behave on highly specialized video domains (e.g., medical imaging, industrial autonomous driving footage) where CLIP's zero-shot vocabulary may fail to extract meaningful semantic cues?
3. The paper utilizes LLaVA to generate captions for the videos. In a real-world pipeline, generating dense captions prior to encoding would introduce massive latency. Is the assumption that captions are pre-existing, or does the total encoding time in Table 3 account for MLLM inference?

**Limitations:**

The authors accurately identify that the framework is highly sensitive to the accuracy of the text prompt. Using irrelevant or missing text causes a significant degradation in both perceptual quality and fidelity, as the model's attention distribution becomes diffuse. While the Template Text fallback is a clever temporary fix, it still underperforms compared to accurate descriptions.

**Strengths And Weaknesses:**

Strengths
1. In table 2, the authors dive deep into where to inject text guidance. The finding that injecting text solely at the encoder (Enc-T/Dec-V) yields the best perceptual quality, cause it shapes the latent space rather than forcing the decoder to hallucinate details from conflicting features.
2. The proposed method is in highly competitive efficiency, requiring only 55.76% of DCVC-FM's per-pixel kMACs while operating faster during inference.
3. The framework consistently achieves state-of-the-art FID and LPIPS scores at low bitrates,  demonstrating that multimodal priors can guide bit allocation toward semantically salient regions.

Weaknesses
1. The model takes a noticeable hit on traditional pixel-level fidelity. The PSNR and MS-SSIM scores drop substantially compared to DCVC-DC and VVC. This gap needs a more robust justification.
2. The reliance on a basic learnable Gain Unit and Inverse Gain Unit for rate adjustment is overly simplistic. The authors acknowledge this as "preliminary," but for a modern end-to-end codec, the lack of continuous, dynamic rate adaptation limits its practical deployment flexibility.
3. There are several distracting errors, for example, in the Figure 3 caption, "compressoin" is misspelled.

---

> ### Author Rebuttal · Authors · 2026-03-31
>
> ## Response
>
> We thank the reviewer for the careful and constructive comments.
>
> ### 1. On the drop in PSNR and MS-SSIM
>
> We agree that our method yields lower PSNR/MS-SSIM in exchange for improved perceptual quality. This reflects a deliberate perception–distortion trade-off rather than an unintended weakness. Our goal is to better preserve semantically important content at low bitrates, rather than optimize distortion-oriented metrics alone. This trade-off is supported by the additional rebuttal evidence, including stronger perceptual and temporal-quality results, as well as the human study.
>
> ### 2. On the simplicity of Gain Unit / Inverse Gain Unit
>
> Thank you for this comment. We agree that the current Gain Unit / Inverse Gain Unit is a simple gain-based design and does not support fully continuous or highly dynamic rate adaptation. In this work, we adopt it as a stable and reproducible solution for multi-rate training across operating points, allowing us to isolate and clearly evaluate the core contribution of text-guided semantic compression. Continuous-rate and content-adaptive rate control are beyond the current scope and remain important future work.
>
> ### 3. On long-term temporal error propagation and flickering
>
> We thank the reviewer for this important concern. To directly examine long-horizon stability, we additionally compressed **384-frame sequences** from UVG (4× longer than the main paper) and report BD-Rate with HM-18.0 as anchor:
>
> |Method|BD-Rate(FID↓)|BD-Rate(LPIPS↓)|BD-Rate(Mean Warping Error↓)|
> |:---|:---:|:---:|:---:|
> |MoVie(Ours)|**-46.90%**|**-81.43%**|**-75.13%**|
> |VTM-23.11|+17.49%|-30.53%|-50.76%|
> |DCVC-FM|-8.09%|+12.46%|-58.32%|
>
> MoVie achieves the best BD-Rate across all three metrics. To further test whether errors accumulate over time, we break down Mean Warping Error BD-Rate into four 96-frame segments:
>
> |Segment|f001–0384|f001–096|f097–192|f193–288|f289–384|
> |:---|:---:|:---:|:---:|:---:|:---:|
> |BD-Rate(Mean Warping Error↓)|-75.13%|-51.66%|-81.94%|-74.99%|-71.91%|
>
> Although the segment-wise results fluctuate, MoVie outperforms HM in all four segments, with no clear progressive degradation over 384 frames. This is consistent with our design, where text acts as conditioning and helps limit semantic drift.
>
> ### 4. On frozen CLIP text encoder and domain generalization
>
> Thank you for this important comment. We agree that frozen CLIP/LLaVA semantics may be less precise in specialized domains, and that our current experiment does not constitute a complete domain-generalization benchmark. As preliminary evidence, we additionally collected ten medical and industrial automation videos and compared perceptual quality with strong baselines:
>
> |Method|bpp|FID↓|FVD↓|LPIPS↓|
> |:--|--:|--:|--:|--:|
> |MoVie|0.0125|**5.8732**|**9.1628**|**0.2317**|
> |DCVC-FM|0.0123|11.8739|15.8342|0.2583|
> |VTM|0.0126|8.6521|13.7631|0.2511|
>
> These results are preliminary, but they suggest that our method remains competitive even when the extracted semantics are less domain-specific.
>
> ### 5. On the practical latency of LLaVA caption generation
>
> Thank you for this important comment. We agree that the latency of LLaVA-based caption generation should be clarified more explicitly.
>
> Table 3 reports only the runtime of the semantic-guided codec and excludes LLaVA caption generation. To quantify this overhead, we measured LLaVA-Video-7B-Qwen2 on MCL-JCV with 96-frame clips, obtaining **12.66 s per clip** (**132 ms per frame**) on average. For clarity, all runtimes in the table below are reported as average per-frame latencies on an NVIDIA A6000 GPU.
>
> |Method|Enc.time|Text.time|Total Enc.time|
> |:--|--:|--:|--:|
> |DCVC-DC|458 ms|–|458 ms|
> |DCVC-FM|439 ms|–|439 ms|
> |VCT|614 ms|–|614 ms|
> |MoVie|324 ms|132 ms|456 ms|
>
> At the same time, semantic guidance does not have to come from online MLLM-generated captions. When videos already have associated text metadata, it can be directly used without extra caption-generation cost. In a preliminary test on ten online videos from YouTube, generated text performs better, but original YouTube text still provides usable semantic guidance:
>
> |Text Source|bpp|PSNR↑|MS-SSIM↑|FID↓|LPIPS↓|
> |:--|--:|--:|--:|--:|--:|
> |Generated Text|0.167|40.469|0.994|5.975|0.103|
> |Original YouTube Text|0.187|39.629|0.989|8.967|0.117|
>
> ### 6. On sensitivity to text quality
>
> We fully agree that our method is sensitive to text quality. Inaccurate, irrelevant, or missing text weakens semantic guidance and can reduce perceptual quality. The template-text fallback is intended only as a robustness-oriented approximation, rather than a substitute for accurate semantics. We will clarify this limitation more explicitly in the revised manuscript and highlight it as an important direction for future work.
>
> ### 7. On writing and presentation issues
>
> Thank you for noting this. We will carefully proofread the manuscript and fix all typographical and formatting issues, including the typo in the Figure 3 caption.

---

> > ### Author Rebuttal · Reviewer_RoSU · 2026-04-04
> >
> > Based on this comprehensive and empirical rebuttal, my concerns have been fully resolved. The authors have defended their methodologies and demonstrated the robustness of their multimodal approach to perceptual video compression. According to the novelty of this paper, I maintain my final recommendation as an Weak Accept.

---

> > > ### Author Response · Authors · 2026-04-07
> > >
> > > Thank you for the thoughtful follow-up and for your positive assessment. We are glad that our rebuttal has adequately addressed your concerns.

---

### Decision · Program_Chairs · 2026-04-30

**Decision:**

Accept (regular)

**Comment:**

Reviewers find the paper technically solid and timely, with a clear practical contribution: integrating text guidance into video compression to improve low-bitrate perceptual quality. Strengths include the video-centric architecture, effective encoder-side text fusion, strong FID/LPIPS results, and competitive efficiency. Concerns remain about weaker pixel-fidelity metrics, incomplete accounting of caption-generation cost, limited temporal-coherence evaluation, and missing stronger video-aware comparisons/metrics. Still, the core idea is novel enough for the area, empirically validated, and likely to stimulate follow-up work on multimodal perceptual codecs.